# Common and divergent gene regulatory networks control injury-induced and developmental neurogenesis in zebrafish retina

Pin Lyu [1,10], Maria Iribarne [2,3,4,10], Dmitri Serjanov[2,3,4,10], Yijie Zhai[1,10], Thanh Hoang[5,6,7,10], Leah J. Campbell[2,3,4], Patrick Boyd[2,3,4], Isabella Palazzo[5], Mikiko Nagashima[8], Nicholas J. Silva [8], Peter F. Hitchcock [8], Jiang Qian [1] ✉, David R. Hyde[2,3,4] ✉ & Seth Blackshaw [1,5,6,7,9] ✉

Following acute retinal damage, zebrafish possess the ability to regenerate all neuronal subtypes through Müller glia (MG) reprogramming and asymmetric cell division that produces a multipotent Müller glia-derived neuronal progenitor cell (MGPC). This raises three key questions. First, do MG reprogram to a developmental retinal progenitor cell (RPC) state? Second, to what extent does regeneration recapitulate retinal development? And finally, does loss of different retinal cell subtypes induce unique MG regeneration responses? We examined these questions by performing single-nuclear and single-cell RNA-Seq and ATAC-Seq in both developing and regenerating retinas. Here we show that injury induces MG to reprogram to a state similar to late-stage RPCs. However, there are major transcriptional differences between MGPCs and RPCs, as well as major transcriptional differences between activated MG and MGPCs when different retinal cell subtypes are damaged. Validation of candidate genes confirmed that loss of different subtypes induces differences in transcription factor gene expression and regeneration outcomes.

The zebrafish retina possesses a remarkable ability to regenerate neurons lost to acute damage by injury-induced reprogramming of Müller glia (MG)[1]. A variety of damage models have been used to study zebrafish retinal regeneration that show varying levels of specificity to neuronal subtypes lost. These include rod and cone photoreceptor ablation using bright light[2–4], excitotoxic and ouabain-mediated destruction of retinal ganglion and amacrine cells[5–7], needle poke destroying all neuronal layers[8], heat damage[9], and the use of cell-specific nitroreductase transgenes to achieve cell type-specific

ablation[5,10–12]. Regardless of the damage model utilized, reprogrammed MG divide asymmetrically to produce a Müller glia-derived neuronal progenitor cell (MGPC), which then continues to proliferate[2,3,13]. These MGPCs regenerate specific cell types that are lost to injury, and these MGPC-derived neural precursors in turn migrate to the appropriate retinal layers[5,10,14,15]. In every retinal injury model, MG rapidly upregulate a subset of genes specific to retinal progenitors such as *ascl1a* and *lin28a* and begin proliferating within 36 h following injury[16–19]. This results in the complete regeneration of

[1]Department of Ophthalmology, Johns Hopkins University School of Medicine, Baltimore, MD 21287, USA. [2]Department of Biological Sciences, University of Notre Dame, Notre Dame, IN 46556, USA. [3]Center for Stem Cells and Regenerative Medicine, University of Notre Dame, Notre Dame, IN 46556, USA. [4]Center for Zebrafish Research, University of Notre Dame, Notre Dame, IN 46556, USA. [5]Department of Neuroscience, Johns Hopkins University School of Medicine, Baltimore, MD 21287, USA. [6]Department of Neurology, Johns Hopkins University School of Medicine, Baltimore, MD 21287, USA. [7]Institute for Cell Engineering, Johns Hopkins University School of Medicine, Baltimore, MD 21287, USA. [8]Department of Ophthalmology and Visual Sciences, University of Michigan School of Medicine, Ann Arbor, MI 48105, USA. [9]Kavli Neuroscience Discovery Institute, Johns Hopkins University School of Medicine, Baltimore, MD 21287, USA. [10]These authors contributed equally: Pin Lyu, Maria Iribarne, Dmitri Serjanov, Yijie Zhai, Thanh Hoang. ✉e-mail: jiang.qian@jhmi.edu; dhyde@nd.edu; sblack@jhmi.edu

all retinal neuronal subtypes within several weeks following initial injury[2,3,15].

Two major models have been proposed to explain regeneration of different neuronal subtypes in the retina. One model proposes that MGPCs respond to the loss of specific neuronal subtypes and are biased to primarily commit and differentiate into those lost neurons in a damage-dependent model[5,14,20,21]. The alternative model suggests that MG reprogram to a state identical to retinal progenitors found in development and the MGPCs then progress through regeneration in a process mimicking retinal development[1,15], with MGPC-derived neuronal subtypes that are not lost to injury presumably failing to integrate efficiently into retinal circuitry and thereafter undergoing apoptosis. Several groups have examined how similar regeneration is to retinal development by analyzing the expression of individual genes that regulate developmental neurogenesis during injury-induced regeneration[14,15,22]. However, these results are limited based on the number of genes and time points examined. In addition, a systematic comparison of the fate of MGPC-derived neurons produced in different injury models has yet to be performed, which is critical for understanding the potential for common features across retinal regeneration.

In this study, we systematically examined the similarities and differences between two different cell-selective models of acute damage: bright light-induced destruction of rod and cone photoreceptors and N-methyl-D-aspartate (NMDA)-mediated excitotoxicity that results in loss of amacrine and ganglion cells. Using long-term 5-ethynyl-2′-deoxyuridine (EdU)-based fate mapping, we found that MGPC-derived neurons generated in these two injury models are not strictly cell type-specific, although they do show a consistent bias toward neurons that are preferentially lost following injury. We found that considerable numbers of inner retinal neurons were generated following light damage, and likewise many photoreceptors following NMDA injury, and these newly generated neurons are not later eliminated by selective cell death.

To investigate the molecular mechanisms controlling injury-induced neurogenesis, we performed both single-cell RNA-Seq (scRNA-Seq), single-nuclei RNA-Seq (snRNA-Seq), and ATAC-Seq multiomic analysis to comprehensively profile injury-induced changes in gene expression and chromatin accessibility seen in both injury models. We sought to determine how closely these MGPCs resembled retinal progenitor cells (RPCs) in developing retinas. We observed that MGPCs produced in both retina damage models exhibit similarities and differences in gene expression and gene regulatory networks that could account for biases in neurogenesis, and we also identified the secreted metalloprotease Mmp9 as a selective inhibitor of amacrine and ganglion cell formation. Though we found that while activated MG reprogrammed into MGPCs whose gene expression profile resemble late-stage RPCs in developing retinas, there were distinct differences that existed between these cell types, with RPCs and MGPCs using distinct gene regulatory networks. We further identified the transcription factor (TF) *foxj1a* as essential for selectively inducing injury-induced neurogenesis. These data demonstrate that retinal regeneration is similar to, but does not precisely recapitulate, retinal development.

## Results

### Light damage and NMDA-induced excitotoxicity result in regeneration of overlapping neuronal cell subtypes

It was previously shown that constant intense light results in the loss of rod and cone photoreceptors[2], while NMDA damage results in the loss of amacrine and ganglion cells[5]. Both damage models induce MG reprogramming, MGPC production and proliferation, and regeneration of retinal neurons. However, a careful analysis of the extent to which different retinal neuronal subtypes are generated in these two injury models has not yet been performed. We labeled proliferating MGPCs using EdU incorporation from 60 to 108 h following both NMDA damage and light damage (LD), examined the neuronal cell subtypes that had incorporated EdU following NMDA damage, and compared these to uninjured controls (Fig. 1a, b). All cell counts are displayed in Source Data. While we detected EdU incorporation in amacrine and ganglion cells at both 7 (7DR) and 14 (14DR) days following NMDA injury, we also observed EdU incorporation into cells throughout both the inner nuclear layer (INL) and the photoreceptor-dominated outer nuclear layer (ONL) at these time points (Fig. 1b, d). No change in either the total number of EdU-positive cells was detected between 7 days of recovery (7DR) and 14DR (Fig. 1c), or in the number of EdU-labeled ONL cells between 7DR and 14DR (Fig. S1a, b). Most EdU-positive cells were found in the INL and ganglion cell layer (GCL) at both time points, with 62.5% of EdU-positive cells at 7DR and 60% of EdU-positive cells at 14DR (Fig. 1e). The remaining EdU-labeled cells were present in the ONL at both time points (Fig. 1e, 37.5 and 40%). Uninjured controls possessed low numbers of EdU-labeled cells in the ONL only, corresponding to adult-born rod photoreceptors as previously reported[5,23,24] (Fig. 1b–d).

Using cell-specific immunohistochemical markers, we observe a slight increase in the number of both EdU/HuC/D-positive amacrine and ganglion cells (Fig. S1a, c) and EdU/Zpr1-positive double cone photoreceptors (Fig. S1f, h) from 7DR to 14DR. In contrast, there is a significant decrease in the number of EdU/PKCα-positive bipolar cells (Fig. S1a, d) and a slight non-significant decrease in the number of EdU/4c12-positive rod photoreceptors (Fig. S1e, g) between 7DR and 14DR. There was no significant difference in the number of EdU/GFAP-positive MG (Fig. S1i, k) and EdU-labeled microglia/macrophages (Fig. S1j, l) from 7DR to 14DR. These results indicate that MGPCs induced by both LD and NMDA primarily generate photoreceptors, amacrine and ganglion cells. Furthermore, with the exception of a small number of bipolar cells, and possibly also rod photoreceptors, no selective loss of any individual subtype of newly generated neurons was detected between 7 and 14 days following NMDA damage, and a large percentage of regenerated neurons were photoreceptors, which were previously thought not to be significantly damaged by NMDA.

We next examined the regenerated retina at 7, 14, and 21 days recovery following light damage, and also observed extensive EdU incorporation in all cell layers (Figs. 1f and S2a). As with the NMDA-injured retina, no change in the total number of EdU-positive cells was detected between 7DR and 21DR (Fig. 1g), or in the total number of EdU-labeled INL cells between (Fig. S2a, b). In contrast to the NMDA-damaged retina, ~60% of all EdU-positive cells were located in the ONL at all time points (58% at 7DR, 63% at 14DR, and 55% at 21DR) (Figs. 1f, h and S2), with ~40% of the EdU-positive nuclei located in the INL and GCL. EdU-positive cells in the ONL were extensively co-labeled with the rod photoreceptor marker 4c12 (Fig. S2a, c), while EdU-positive cells in the INL were predominantly HuC/D-positive (Figs. 1l and S2e, f). As with NMDA injury, few EdU-positive cells colabeled with either the MG marker GFAP (Fig. S2a, d) or the bipolar cell marker PKCα (Fig. S2e, g), while essentially no EdU-positive horizontal cells were detected. Again, the total number of EdU-labeled cells remained statistically the same for all cell type-specific markers tested from 7DR through 21DR (Figs. 1g and S2), demonstrating that there was no selective loss of any particular neuronal subtype in this period.

To examine the reason for this widespread EdU incorporation in both damage models, we carefully examined the loss of retinal neurons in both damage models. During NMDA damage, we quantified the number of DAPI-stained nuclei in all three nuclear layers in undamaged (control) and NMDA-damaged retinas at 48, 60, and 72 h following NMDA injection (Fig. 1i, j). As expected, there is a significant reduction in the number of GCL nuclei relative to control, but not a significant reduction in the number of INL nuclei (Fig. 1j). However, there is also an unexpected and significant reduction in the number of ONL nuclei at

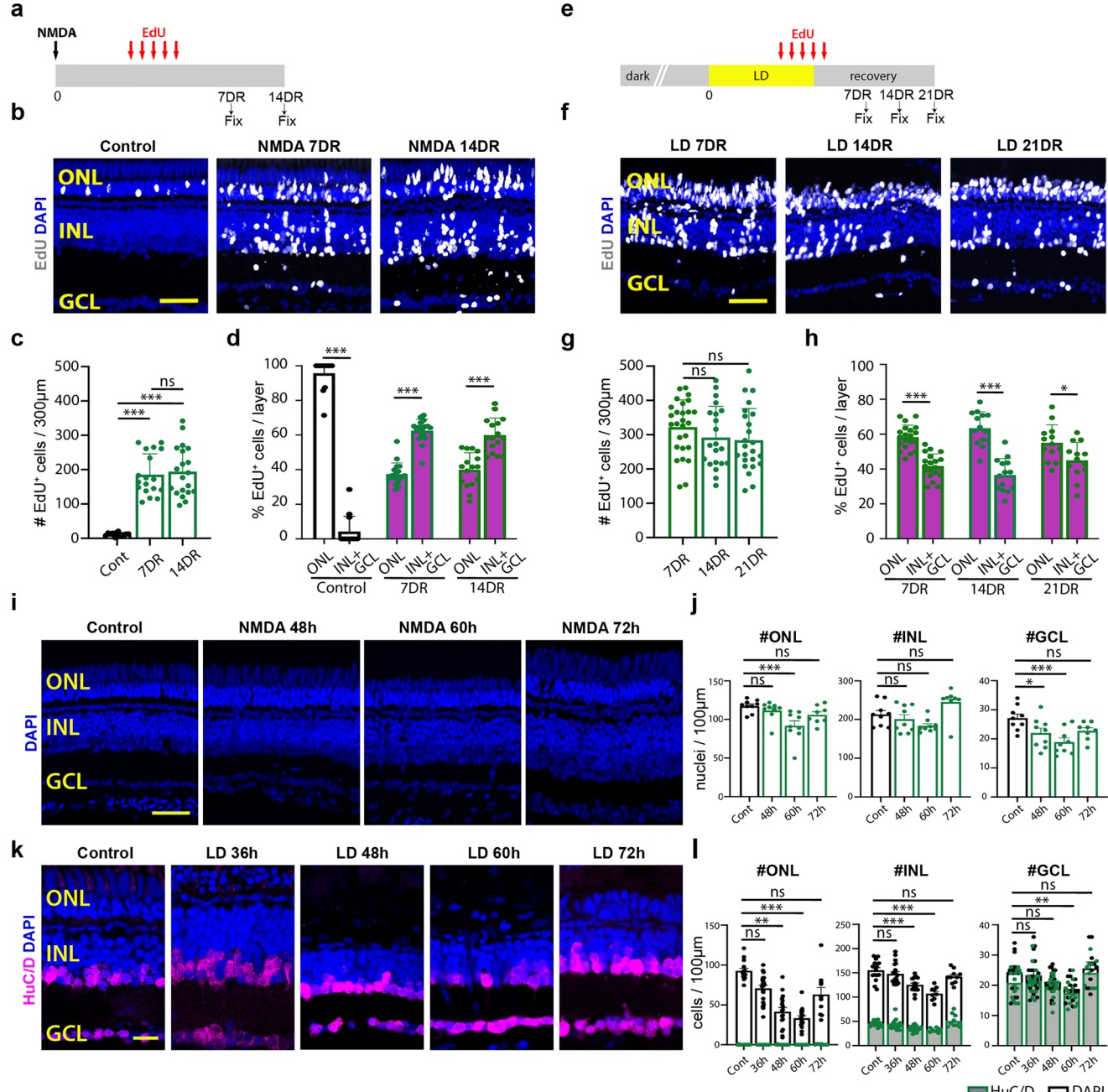

**Fig. 1 | Comparison of NMDA-induced and light-induced retinal damage.**
**a** Schematic of NMDA-induced damage experiment. **b** EdU-labeling in control retinas and following NMDA damage. **c** Quantification of the number of EdU-labeled cells in all three retinal layers. Control (Cont) $n = 13$, 7 days recovery (DR) $n = 18$, 14DR $n = 20$. Three independent experiments. **d** The percentage of EdU-labeled cells in the Outer Nuclear Layer (ONL) vs. combined in the Inner Nuclear Layer (INL) and Ganglion Cell Layer (GCL). Cont ONL and INL + GCL $n = 13$; 7DR ONL and INL + GCL $n = 18$, 14DR ONL and INL + GCL $n = 20$. Three independent experiments. **e** Schematic of light-induced damage (LD) experiment. **f** EdU-labeling following light damage (LD). **g** Quantification of the number of EdU-labeled cells in all three retinal layers. 7DR $n = 27$, 14DR $n = 21$, 21DR $n = 24$. Three independent experiments. **h** The percentage of the EdU-labeled cells in the ONL vs. combined in the INL + GCL. 7DR ONL and INL + GCL $n = 27$; 14DR ONL and INL + GCL $n = 24$; 21DR ONL and INL + GCL $n = 24$. Three independent experiments. **i** DAPI staining of undamaged retinas and 48, 60, and 72 h after injecting NMDA. **j** Quantification of the number of DAPI-labeled nuclei in the ONL, INL, and GCL. Cont, 48 h, and 60 h (ONL, INL, GCL) $n = 9$; 72 h (ONL, INL, GCL) $n = 8$. Three independent experiments. **k** DAPI and HuC/D staining of undamaged retinas and 36, 48, 60, and 72 h after starting constant light treatment. **l** Quantification of the number of DAPI- or HuC/D-labeled nuclei in the ONL, INL, and GCL. Cont (ONL, INL, GCL) $n = 15$; 36 h (ONL, INL, GCL) $n = 20$; 48 h (ONL, INL, GCL) $n = 17$; 60 h (ONL, INL, GCL) $n = 16$; 72 h (ONL, INL, GCL) $n = 10$. Scale bars in (**b**, **f**, **i**) are 20 μm and in (**k**) is 14 μm. **c**–**h**, **j**, **l** Data are presented as mean values ± SEM. For statistical analysis, one-way ANOVA was followed by $t$-test with Dunnett's method for multiple comparisons correction. Asterisks indicate statistically significant differences between the indicated groups (*$p \le 0.05$, **$p \le 0.01$, ***$p \le 0.001$). Source data are provided as a Source Data file 1.

60 h following NMDA injection relative to the control (Fig. 1j). Likewise, in light-damaged retinas, a significant loss of ONL nuclei was detected by 36 h, with lowest numbers seen at 60 h, and a recovery to baseline levels observed by 72 h post-injury (Fig. 1k, l). However, we also observed significant reductions in the number of nuclei in both the INL

and GCL by 48 and 60 h following light damage, respectively (Fig. 1k, l). The rapid recovery seen in the number of DAPI-positive neurons may represent both newly-generated MGPC-derived neurons and, particularly in the ONL, the presence of MGPC cells undergoing interkinetic nuclear migration[25].

Analysis of light-damaged retinas showed that while TUNEL-positive and pyknotic cells are largely restricted to the ONL at 24 h following damage, at 48 h post-injury substantial numbers of pyknotic cells are also observed in the INL and GCL (Fig. S3a). Likewise, while dying cells are restricted to the inner retina at 24 h following NMDA treatment, they are also observed in the ONL at 48 h post-injury (Fig. S3b). Both light damage and NMDA-mediated excitotoxicity therefore result in the loss of unexpected retinal cell types (photoreceptors and inner retinal neurons, respectively) that likely account for the presence of regenerated neurons in those layers.

## ScRNA-Seq and snRNA/ATAC-Seq multiomic analysis of NMDA- and light-damaged retina

To comprehensively characterize changes in gene expression and regulation that occur following both NMDA and light-mediated injury, we performed both scRNA-Seq and combined multiomic snRNA/ATAC-Seq analysis of the whole retina at multiple time points following injury. We analyzed uninjured control samples along with retinas harvested at 36, 54, 72, 96 h, 7 days, and 14 days post-injury (Supplementary Dataset 1). In total, we profiled 99,555 cells using scRNA-Seq and 137,490/127,364 cells using snRNA/ATAC-Seq multiomic analysis. UMAP analysis of integrated datasets from both scRNA-Seq and snRNA/ATAC-Seq multiomic analysis readily identified all major adult retinal cells in both uninjured control and each condition examined (Figs. 2a, b and S4a, b). In addition, specific cell subtypes were detected in both injury models that were either absent or present at much lower levels in uninjured controls. These include activated MG, proliferating MGPCs and immature postmitotic precursors of rod and cone photoreceptors, amacrine and ganglion cells (Fig. 2c). More rod and cone postmitotic precursors were consistently detected following light damage, while more RGC postmitotic precursors were detected following NMDA treatment (Figs. 2d and S4c). Relatively few bipolar and no horizontal cell postmitotic precursors were detected, matching the observed EdU data in Fig. 1. Well-characterized cell type-specific markers showed both expected patterns of gene expression and chromatin accessibility as determined by both scRNA-Seq and snRNA/ATAC-Seq multiomic analysis (Fig. S4d and Supplementary Dataset 2: T1–T4). A full list of selective markers for each cell cluster observed in the scRNA- and snRNA-Seq datasets is included in Supplementary Dataset 3.

Analyzing the integrated snRNA/ATAC-Seq dataset (Fig. 2a), we observed dynamic changes in cell composition across each time point for both injury models (Fig. 2b–d). We noticed a progressive reduction in the relative fraction of activated MG and MGPCs from 36 h onward, while the majority of MG returned to a resting state by 14 days following injury (Fig. 2d). A modest but significant increase in the relative fraction of activated MG was observed following NMDA injury relative to light damage (Fig. 2d). The fraction of RGC precursors likewise decreases from 36 h following NMDA injury but declines more slowly following light damage, while the fractions of amacrine and cone photoreceptor precursors peak at 96 h following both NMDA and light damage (Fig. 2d). Finally, the fraction of rod photoreceptor and bipolar precursors steadily increase with time for both injury models (Fig. 2d). This broadly corresponds to the observed order of neurogenesis in the developing zebrafish retina, where ganglion cells are born first and bipolar cells and rod photoreceptors last[26,27], and also aligns with results obtained from a recent scRNA-Seq analysis of MGPC and MGPC-derived neurons obtained from light-damaged retina[28].

To identify genes that control both injury-induced MG reprogramming and specification and differentiation of MGPC-derived neurons, we performed pseudotime analysis of the integrated snRNA/ATAC-Seq dataset from both NMDA and light damage samples (Fig. 2c). We identified six distinct trajectories, corresponding respectively to the transition from resting MG to activated MG to MGPCs; from MGPCs to RGCs, amacrine cells, bipolar cells, cone

photoreceptors, and rod photoreceptors (Fig. 2c). Dynamic patterns of differential accessibility, which were detected using scATAC-Seq analysis, were observed for both promoter and putative enhancer elements (Fig. 2e). A total of 4891 common dynamically expressed genes and 13,988 common differentially accessible DNA elements were observed in both NMDA and light-damaged samples (Supplementary Dataset 4, T1, T2).

Within the first trajectory, three separate sequential elements were detected, corresponding to genes selectively expressed in resting MG, activated MG, and MGPCs respectively (Fig. 2e and Supplementary Dataset 4: T1, T2). Resting MG expressed TFs known to mediate glial quiescence such as *nfia*, activated MG expressed TFs such as *vsx2* and *foxj1a*, while MGPCs expressed *atoh7* and cell cycle regulators such as *mki67*. Each trajectory included multiple TFs previously shown to be essential for specification of the respective cell type *pou4f1* in RGCs (2), *tfap2c/d* in amacrine cells (3), *vsx1* in bipolar cells (4), *nrl* in rods (5), and *six7* in cone photoreceptors (6). We also analyzed the scATAC-Seq data for differentially accessible TF binding motif sequences, and identified 184 common differentially accessible motifs (Fig. 2f and Supplementary Dataset 4: T3). These likewise include target sites for many TFs that are differentially expressed in either NMDA or light damage, including *foxj1a*, *tfap2b*, and *nrl* (Fig. 2g, h). Motifs for TFs that are not differentially expressed but which promote cone photoreceptor specification, such as *onecut1*[29,30], are also enriched following light damage (Fig. 2h).

## NMDA and light damage differentially regulate gene expression in MG and MGPCs

These data show that while NMDA-mediated excitotoxicity and light damage induce formation of MGPCs through broadly overlapping mechanisms, there are nonetheless important quantitative differences seen in both gene expression and regulation in these two injury models. To identify differences in gene expression and regulation in MG following NMDA and light damage, we directly compared differences in mRNA expression, and both chromatin and motif accessibility in the two injury models. While we identified 949, 461, and 628 genes that showed similar patterns of expression in both light damage and following NMDA treatment, we also identified 434, 307, and 437 genes showing higher expression in light damage and 397, 216, and 319 genes following NMDA (Fig. 2g and Supplementary Dataset 4: T1 and T5) in MGs, activated MG, and MGPCs. These include several genes that have been previously investigated in the context of developmental and/or injury-induced neurogenesis[31–36]. Specifically, we observe *bmpr1ab*, *metrn*, *crtac1a*, and *cxcl12b* selectively and rapidly upregulated in activated MG of light-damaged retina, while *vegfaa/b*, *stat1a/b*, *stat2*, *cxcl14*, *her9*, *fstl1a*, and *mmp9* are rapidly upregulated following NMDA treatment (Fig. 2g and Supplementary Dataset 4: T5). We also observe more pronounced upregulation of *foxj1a*, a TF which directs formation of multiciliated epithelial cells[37,38], as well as the transcription factor *tbx4*, following light damage. Both Gene Ontology and KEGG pathway analysis identified functional differences in differentially expressed genes between the two injury conditions, with genes associated with cell morphogenesis and adhesion enriched following light damage and genes controlling stress and immune response, as well as amino acid biosynthesis, enriched following NMDA treatment (Fig. S5a, b).

We likewise identified 3833, 1267, and 2051 differential genes related chromatin regions that showed similar patterns of differential accessibility in both injury models, 165, 83, and 240 chromatin regions that showed higher expression following light damage and 216, 141, and 206 chromatin regions that showed higher expression following NMDA treatment in the snATAC-Seq dataset in MGs, activated MG, and MGPCs (Fig. 2g and Supplementary Dataset 3: T2 and T6). Analysis of differentially accessible motifs revealed that 33, 30, and 35 motifs showed shared patterns of differential accessibility in both injury models, but that 8, 2, and 9 motifs were selectively enriched following

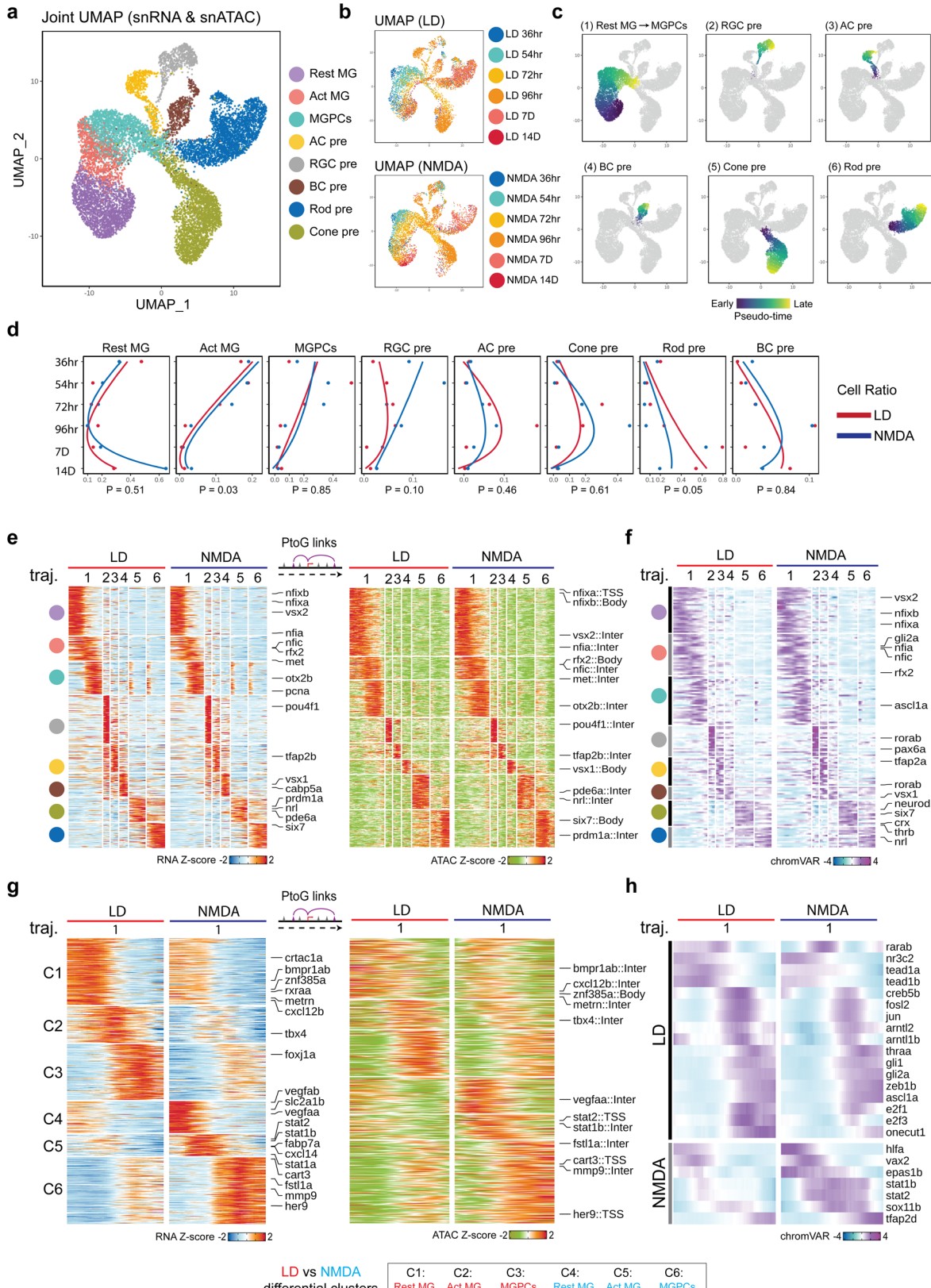

light damage in MGs, activated MG, and MGPCs, while 3, 4, and 4 motifs were enriched following NMDA treatment, with most showing relatively modest differences. Light damage induced increased accessibility at *tead1*, *nr2e1*, *rxraa*, and *arntl1b* motifs, while NMDA induced increased accessibility at *stat1a/b*, *stat2*, *rfx2*, and *vax2* motifs (Fig. 2h and Supplementary Dataset 4: T3 and T7).

## Mmp9 selectively suppresses regeneration of amacrine and ganglion cells

It has been previously shown that Mmp9 protein acts to repress MGPC proliferation in the light-damaged retina, with the light-damaged *mmp9* mutant exhibiting an increased number of MGPCs relative to light-damaged controls, and also exhibiting a loss of regenerated

**Fig. 2 | Shared and differential patterns of gene expression and chromatin accessibility data observed in MG-derived cells following LD and NMDA treatment. a**, **b** Combined UMAP projection of Müller glia (MG) and progenitor neuron cells profiled using multiomic sequencing. Each point (cell) is colored by cell type (**a**) and time points (**b**). Resting (Rest); Activated (Act); MG-derived progenitor cells (MGPCs); amacrine cell (AC); retinal ganglion cell (RGC); bipolar cell (BC); precursor (pre). **c** UMAPs showing trajectories constructed from mutiomics datasets of combined light damage (LD) and NMDA datasets. Color indicates pseudotime state. **d** Line graphs showing the fraction of cells (*x* axis) at each time point (*y* axis) of each cell type. Lines are colored by treatment. The paired *t*-test were used to compare the cell fraction from LD and NMDA. *p* values are labeled at the bottom of the graph. **e** Heatmap shows the consensus marker genes and their related marker peaks (TSS and enhancer) between LD and NMDA treatment for each cell type. **f** Heatmap shows the consensus motifs between LD and NMDA treatment for each cell type. **g** Heatmap shows the differential genes and their related differential peaks (TSS and enhancer) between LD and NMDA treatment for MG (Rest), MG (Act) and MGPCs. **h** Heatmap shows the differential motifs between LD and NMDA treatment for MG (Rest), MG(Act), and MGPCs.

cones relative to control retinas[36]. However, we observed that *mmp9* expression was elevated in activated MG and MGPCs in the NMDA-damaged relative to the light-damaged retina (Fig. 3a), and that NMDA treatment led to increased accessibility at putative cis-regulatory elements regulating *mmp9* transcription (Fig. 3b). This suggested that Mmp9 may preferentially modulate the generation of inner retinal neurons, and we therefore examined the consequence of loss of *mmp9* expression in the regenerated retina in the two damage models, comparing the total number of EdU-labeled cells in undamaged control, NMDA-damaged, and light-damaged retinas at 7DR, and visualizing amacrine and ganglion cells with antibodies against HuC/D (Fig. 3c–g).

We observed a significant increase in the total number of EdU-labeled neurons in the *mmp9* mutant relative to controls following light damage (Fig. 3d), as previously reported[36], which reflected an increase in the number of EdU-positive cells in both the ONL and INL (Fig. S6c, d). However, there was no significant increase in the total number of EdU-labeled neurons in the *mmp9* mutant following NMDA damage relative to controls (Figs. 3d and S6a, b). We also observed a significantly greater number of EdU-positive cells in both the INL and GCL in undamaged retinas, when the vast majority of newborn neurons are rod photoreceptors[27], as well as following light damage (Fig. 3e). However, this was not observed in NMDA-damaged retinas. Following both light damage and NMDA treatment, moreover, we observe a significant increase in both the relative ratio of EdU-positive cells in the INL and GCL relative to the ONL (Fig. 3e), as well as an absolute increase in the number of HuC/D and EdU-positive amacrine and ganglion cells (Fig. 3f, g). All cell counts are listed in Source Data.

The relative increase in generation of inner retinal neurons is already evident at 3DR in *mmp9* mutants following light damage, though not following NMDA treatment (Fig. S6e–j), possibly indicating the existence of additional factors inhibiting the generation of inner retinal neurons that initially compensate for loss of function of *mmp9*. The increased number of regenerated HuC/D-positive amacrine and ganglion cells in *mmp9* mutants relative to controls suggests that Mmp9 actively represses specification and/or differentiation of MGPC-derived amacrine and ganglion cells, in addition to its previously known role in repressing MGPC proliferation. To our knowledge, the *mmp9* mutant represents the first mutant that alters the commitment of MGPCs from one neuronal subtype (photoreceptors) to another subtype (amacrine and ganglion cells).

### Identification of gene regulatory networks controlling differential injury response in NMDA and light-damaged MG

Having observed extensive differences in both gene expression and chromatin accessibility in activated MG and MGPCs following NMDA and light damage, we hypothesized that injury-induced gene regulatory networks are differentially organized in these two injury models (Fig. 4a). To test this, we used gene expression and chromatin accessibility data extracted from multiomic data to identify key transcriptional activators and repressors, finding that 73.6 and 69.2% of identified TF-Motif pairs act as transcription activators in LD and NMDA injury models (Fig. 4b and Supplementary Dataset 3: T1). We next separately reconstructed injury-regulated transcriptional

regulatory networks in MG and MGPCs following both NMDA treatment and light damage, identifying both TFs and target genes that directly target enhancer and/or promoter elements to activate or repress individual genes (Fig. 4c). We identified far more candidate activating relationships than repressive regulatory relationships in both injury models, with 86.3%/86.4% of these targeting enhancers in light damage and NMDA treatment respectively, and 13.7%/13.6% targeting promoters (Fig. 4c). We observe that 28.9%/34.1% of all TF-peak-gene and 42.1%/46.8% of all TF-gene regulatory relationships are common between the two injury models (Fig. 4d).

We next identified transcriptional regulatory relationships that were specifically active in either light-damaged or NMDA-treated samples. While light damage-specific regulons were predominantly activating—just like injury-induced regulons as a whole—the number of NMDA-specific regulons was much much smaller, and contained roughly equal numbers of activating and repressive regulons (Fig. 4e and Supplementary Dataset 3). To identify key transcription factors (TFs) in specific regulons for each model, we ranked TFs based on their specificity scores and the number of their target-specific genes. These values are determined by the total regulons and specific regulons associated with each individual. Light damage-specific activating regulons were selectively targeted by *ascl1a*, *gli2a*, *lhx2b*, *rarab*, *rarga*, *rxraa*, *sox13*, and *zic2b*. In contrast, NMDA-specific activating regulons were heavily targeted by multiple Stat factors, including *stat1a*, *stat1b*, and *stat2*, with these factors also activating one another's transcription (Fig. 4f). Several factors in this network—including *gli3* and *sox11b*—were also predicted to directly activate *mmp9* transcription, while *atoh7* promotes RGC specification and survival[39,40]. Light damage-specific negative regulons were heavily targeted by *hmga1a* and *sox11b*, which function primarily as activators in NMDA-treated retina (Supplementary Dataset 3: T2, T3).

Analysis of both scRNA-Seq and snRNA-Seq data also identified a cell cluster corresponding to microglia and/or macrophages, as indicated selective expression of markers such as *ptprc* (*cd45*), *mpeg1.1*, and *csf1ra* (Fig. S3 and Supplementary Dataset 5). By analyzing scRNA-Seq data obtained from microglia and/or macrophages in both light damaged and NMDA-treated retina, we observed that *il1b*, *tnfsf12*, *stat1a/b*, and *stat2* were all selectively upregulated following NMDA treatment (Fig. 4g), indicating a potential role for differential microglial activation in regulating injury-specific patterns of MG reprogramming. Interestingly, *stat1a/b* and *tnfsf12* are also transiently downregulated following light damage, although the reason for this is unclear. Finally, analysis of activating TFs specific to both injury models showed that while many individual TFs were selectively active in both MG and MGPCs following light damage, the smaller number of activating TFs seen following NMDA treatment was generally specific to individual cell clusters, with Stat factors selectively active in activated MG, and *atoh7* active in MGPCs (Fig. 4h).

### Differential organization of gene regulatory networks controlling injury-induced and developmental neurogenesis

To further investigate similarities and differences between gene regulatory networks (GRNs) controlling injury-induced and

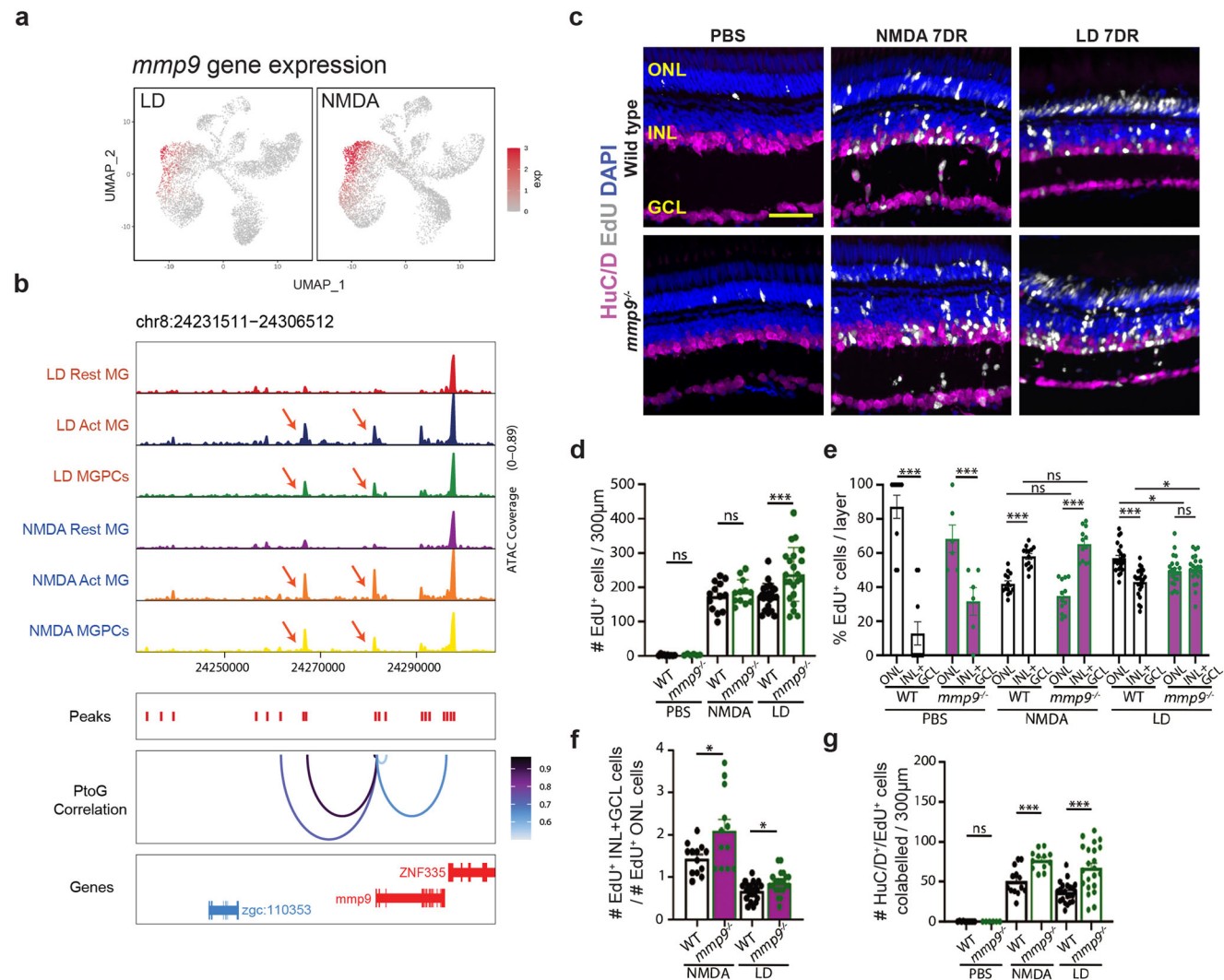

**Fig. 3 | Mmp9 selectively inhibits generation of inner retinal neurons from MGPCs. a** UMAP plot showing expression pattern of *mmp9* in LD (light damage) and NMDA-treated retina. **b** Altered chromatin accessibility in putative regulatory sequences associated with *mmp9* following both LD and NMDA damage. **c** PBS-injected control, NMDA-treated, and LD retinas at 7 days recovery (DR) immunostained with HuC/D to label retinal ganglion cells and amacrine cells, EdU, and counterstained with DAPI. Outer nuclear layer (ONL); inner nuclear layer (INL); ganglion cell layer (GCL). **d** Quantification of the number of EdU-labeled cells in all three retinal layers in wild-type (WT) and *mmp9* mutants following either PBS injection, NMDA damage, or LD. (PBS) WT $n = 10$, *mmp9*$^{-/-}$ $n = 6$; (NMDA) WT $n = 13$, *mmp9*$^{-/-}$ $n = 12$; (LD) WT $n = 23$, *mmp9*$^{-/-}$ $n = 22$. Three replicate experiments. **e** The percentage of EdU-positive cells in the ONL vs. the INL + GCL is plotted for wild-type and *mmp9* mutant retinas after PBS injection, NMDA damage, and light damage.

(PBS) WT $n = 10$, *mmp9*$^{-/-}$ $n = 6$; (NMDA) WT $n = 13$, *mmp9*$^{-/-}$ $n = 12$; (LD) WT $n = 23$, *mmp9*$^{-/-}$ $n = 22$. Three replicate experiments. **f** The ratio of EdU-positive ONL cells to EdU-positive INL + GCL cells is plotted for wild-type and *mmp9* mutant fish following either NMDA damage or light damage. (NMDA) WT $n = 13$, *mmp9*$^{-/-}$ $n = 12$; (LD) WT $n = 23$, *mmp9*$^{-/-}$ $n = 22$. Three replicate experiments. **g** Quantification of the number of cells colabelled for EdU and HuC/D in PBS-injected, NMDA-injected, and light-damaged retinas. (PBS) WT $n = 10$, *mmp9*$^{-/-}$ $n = 6$; (NMDA) WT $n = 13$, *mmp9*$^{-/-}$ $n = 12$; (LD) WT $n = 23$, *mmp9*$^{-/-}$ $n = 22$. Three replicate experiments. Magenta bars indicate data from *mmp9* mutants. Scale bar in (**c**) is 20 µm. **d**–**g** Data are presented as mean values ± SEM. Students' *t* test was performed for (**d**, **f**, **g**), while two-way ANOVA with Bonferroni's post hoc test for (**e**). Asterisks indicate statistically significant differences between the indicated groups (*$p \leq 0.05$, ***$p \leq 0.001$). Source data are provided as a Source Data file 1.

developmental neurogenesis, we generated multiomic snRNA-Seq/ATAC-Seq from developing zebrafish retina, profiling the full time course of developmental neurogenesis. We profiled whole embryo heads from three early time points (24hpf, 30hpf, and 36hpf) at which eyes could not be cleanly dissected, and whole retina from six later time points (48hpf, 54hpf, 60hpf, 72hpf, 4dpf, and 5dpf), profiling a total of 52,695 cells (Supplementary Dataset 5). Integrated UMAP analysis of this dataset was able to clearly distinguish separate populations of early- and late-stage RPCs, as well as clusters corresponding to each major retinal cell type (Fig. S7a). We observed that the relative fraction of early-stage RPCs diminished rapidly between 36 and 48hpf, while the fraction of late-stage RPCs peaked at 54hpf and declined rapidly thereafter. As expected, RGCs are the first

postmitotic cells to be detected, followed rapidly thereafter by horizontal and amacrine cells and cone photoreceptor precursors. Similar results were observed following reanalysis of previously published scRNA-Seq data obtained from developing zebrafish retina (Fig. S7b)[41]. The relative fraction of mature cones, rods, bipolars and Müller glia increases rapidly after 72hpf (Fig. S7c). As is the case in the injured adult retina, each cell cluster in both the multiomic and scRNA-Seq datasets is readily delineated by enriched expression and/or greater chromatin accessibility associated with well-characterized marker genes (Fig. S7d).

To directly compare GRNs controlling the progression of neurogenesis in injured and developing retina, we integrated the two datasets, projecting them into a common UMAP space (Fig. 5a). Both

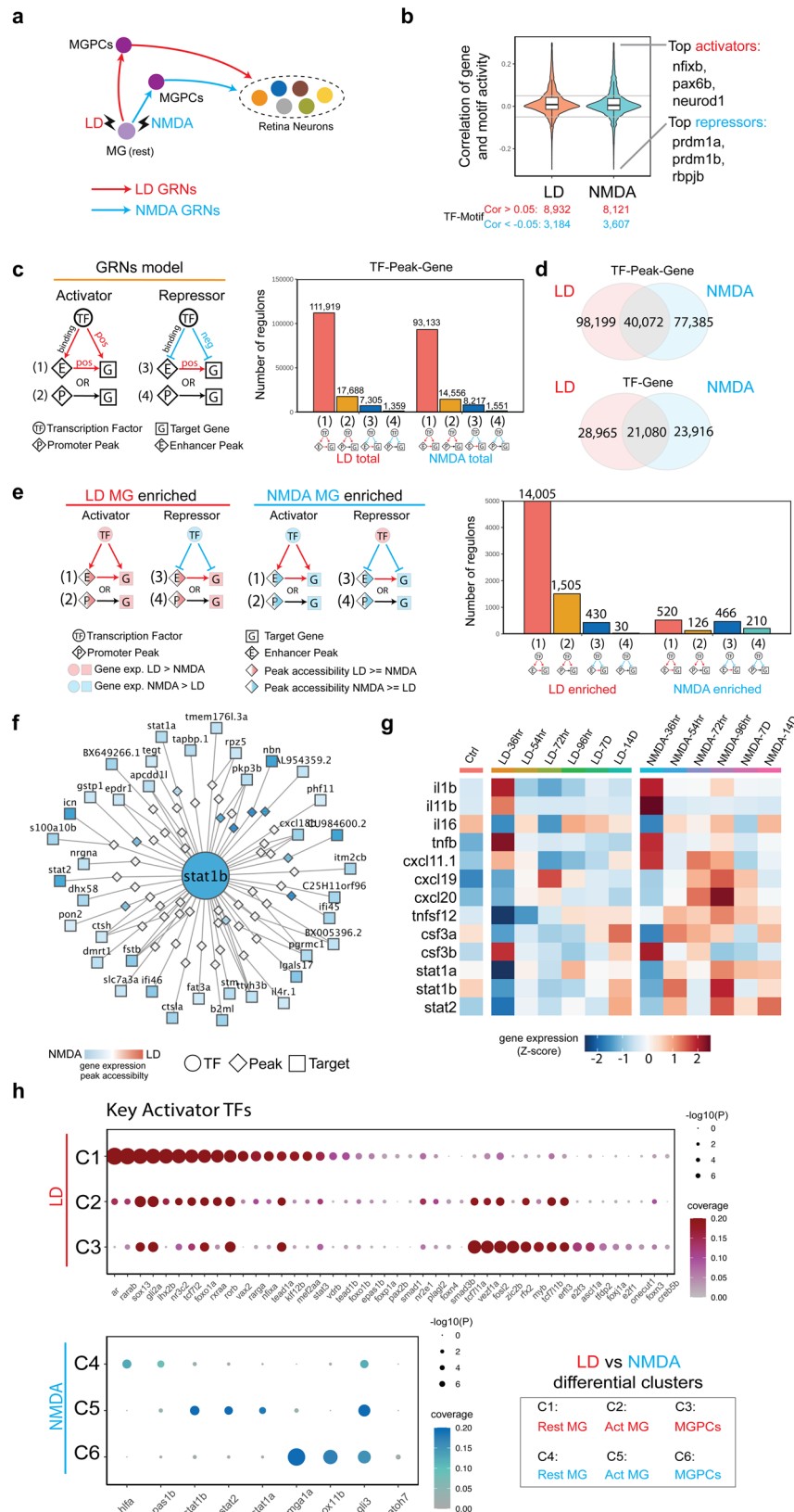

multiomic and scRNA-Seq data detected distinct clusters corresponding to early and late-stage RPCs, as well as immature precursors corresponding to every major retinal cell type (Fig. S7a, b). These cell clusters showed expected changes over developmental time, with early-stage RPCs absent by 60hpf, late-stage RPCs greatly reduced in abundance by 5dpf, and small numbers of RGCs

detectable by as early at 24hpf, in line with previous observations[42] (Fig. S7c and Supplementary Dataset 5). Small numbers of amacrine and horizontal cells were detected shortly thereafter, with bipolar cell and Müller glia not evident till 60hpf. Although immature photoreceptor precursors were detected as early as 48hpf, large numbers of rods and cones were not observed until 72hfp in the

**Fig. 4 | Transcription factors controlling differential gene expression in MG and MGPC following LD and NMDA treatment. a** Schematic of gene regulatory networks (GRNs) that direct Müller glia (MG) reprogramming and MG-derived progenitor cell (MGPC) differentiation after injury via light damage (LD) and NMDA treatment. **b** Inference of activator and repressor function for each individual transcription factor from multiomic datasets. The *y*-axis represents the correlation distribution between gene expression and chromVAR score. The top three activator and repressor TF-Motif pairs are shown on the right. The center, lower/upper bound of the boxplot shows the median, 25th and 75th of the correlations, The whiskers extend from the ends of the box to the smallest and largest values that are within 1.5 times the IQR from the lower and upper quartiles, respectively. **c** Gene regulatory networks of LD and NMDA datasets. (left) Triple regulons model. A circle indicates a TF, a rectangle indicates a target gene, and a peak. (right) barplot shows the types of regulons between LD and NMDA datasets. **d** Venn diagram shows the overlap of regulons between LD and NMDA datasets. **e** Enriched gene regulatory networks of LD and NMDA treatment. (left) Enriched Triple

regulons model for each condition. A circle indicates a TF, a rectangle indicates a target gene, and a diamond indicates a peak. Color indicates the log2 fold change of gene/peak between LD and NMDA datasets.(right) barplot shows the number of different types of regulons between LD and NMDA enriched GRNs. **f** An example of stat2 regulons. A circle indicates a TF, a rectangle indicates a target gene, and a diamond indicates a peak. Color indicates the log2 fold change of gene/peak between LD and NMDA datasets. **g** Heatmap showing the differentially expressed genes in microglia/macrophages after LD and NMDA treatment. Microglia/macrophage cells are ordered by time points after injury, and an averaged expression level is shown for each time point. Color represents mean-centered normalized expression levels. **h** Dotplot displays the key activator TFs for each divergent gene cluster. The color represents the ratio of the TF's targets within that gene cluster, while the dot size indicates the *p* value of the TF's regulatory specificity for the respective gene cluster. Hypergeometric test were used test whether the TF's targets are enriched in the DEG clusters.

multiome data, and they could not be resolved in the scRNA-Seq data.

All identified clusters in both datasets expressed well-characterized molecular markers (Fig. S7d). We observed that gene expression patterns in MG from the developmental dataset correlated well between both resting and activated MG from the injury dataset, while MGPCs correlated well with data from late-stage RPCs. We did not, however, observe a distinct early-stage RPC-like cluster in the injury dataset. Immature precursors of each major neuronal cell type likewise correlated most closely with the corresponding cell type present in developing retina, although horizontal cell precursors were absent from the injury dataset (Fig. 5c, d).

We inferred a common trajectory from resting MG to activated MG to MGPCs on the one hand, and from MG to RPCs on the other (Fig. 5b) based on the combined UMAPs. While we observed extensive similarities between gene expression profiles, chromatin accessibility patterns, and TF motif accessibility in both datasets (Fig. 5e, f and Supplementary Dataset 6: T1−T4), we also observed a large number of differentially expressed genes in both trajectories (Fig. 5g, h). Injury-induced trajectory enriched for genes expressed at high levels in mature resting MG (*her6, notch2*) and for genes selectively expressed in activated MG (*mmp9, clcf1, met, foxj1a, lin28a*) (Fig. 5g and Supplementary Dataset 6: T5). Gene Ontology and KEGG analysis shows that genes associated with cell adhesion, neurogenesis, and FoxO and VEGF signaling are enriched in the injury dataset, while genes controlling cell cycle, protein synthesis and splicing were enriched in the developmental dataset (Fig. S5c, d). We likewise observed major differences in both differential chromatin accessibility (Fig. 5g) and TF motifs (Fig. 5h) between the injury and developmental datasets. Notably, we observe that the promoters of many genes selectively expressed in resting and/or activated MG−including *her6, met, mmp9, lin28a*, and *foxj1a*−are not accessible in RPCs in the developing retina (Fig. 5g). We likewise observe that target motifs for some of these same factors, including *stat3* and *foxj1a*, are likewise accessible only in the combined injury dataset (Fig. 5h).

### Identification of gene regulatory networks differentially regulating neurogenesis in MGPCs and RPCs

Having observed extensive similarities and differences in both gene expression and chromatin accessibility over the course of neurogenesis in MGPCs and RPCs, we next reconstructed and directly compared dynamic GRNs in both cell types using the same methods we previously used to compare GRN in NMDA and light-damaged MG and MGPCs (Fig. 6a). The aggregated light damage and NMDA injury-regulated GRNs consisted of 11,925 TF-Motif pairs, with developmental TFs comprising 10,954 of these, with a 70.3% and 71.9% of activators respectively (Fig. 6b and Supplementary Dataset 7). Analysis of injury and development-associated regulons showed that 91.6% of injury-

associated and 97.0% of development-associated regulons were activating (Fig. 6c). 6.9%/11.7% of development-specific TF-peak-gene interactions and 13.6%/23.5% of TF-gene interactions overlap with the injury-specific dataset (Fig. 6d). Analysis of injury and development-specific regulons reveals that these are likewise dominated by activating relationships, with more than five times as many total injury-specific regulatory relationships than developmental-specific regulatory relationships (Fig. 6e).

Development-enriched activating TFs identified using this analysis include *hmga1a, foxp4, pax2a, rx1*, and *sox3*. Injury-enriched activating TFs include *ascl1a, rfx2, sox2, sox13, stat3, tead1a*, and *zic2b* (Fig. 6g and Supplementary Dataset 7). An example of this category of *a* key injury-induced regulatory TF is shown for *foxj1a*, which is selectively expressed in the regenerating retina (Fig. 6f). We observe that *foxj1a* directly regulates several genes that regulate MGPC formation, such as *mmp9, foxn4*, and *sox2*[18,36,43]. Notably, *foxj1a* is also more strongly induced following light damage than NMDA treatment (Fig. 2g), indicating that it may also differentially regulate injury-induced neurogenesis in these two models. We observe that many of the injury-enriched activating TFs are strongly and selectively expressed in activated MG, which are absent from the developmental dataset, while development-enriched TFs are predominantly active in early-stage RPCs, which are likewise lacking from the injury-induced dataset (Fig. 6g).

### *Foxj1a* is necessary for neuronal regeneration of the damaged adult retina, but it is not required for retinal neurogenesis

To determine if any of these candidate injury-induced TFs are required for neuronal regeneration, but not in retinal development, we tested *foxj1a* (Figs. 5g, h and 6f). *foxj1a* is expressed much higher during retinal regeneration than retinal development (Fig. 7a, b). To test the role of *foxj1a* in regeneration, morpholinos were intravitreally injected and electroporated into either NMDA-damaged or light-damaged retinas at the start of damage[44]. As controls, we used the Standard Control morpholino, which is not complementary to any known sequence in the zebrafish genome (GeneTools, LLC), and the *pcna* (*proliferating cell nuclear antigen*) morpholino, which is necessary for MG and MGPC proliferation during retinal regeneration[45]. As expected, the *pcna* morphant retina possessed significantly fewer PCNA-positive proliferating MGPCs at 72 h following either NMDA damage (Fig. 7c−e) or light damage (Fig. 7f−h). We likewise observed significantly reduced levels of PCNA+ cells in the ONL and INL of both NMDA-treated and light damaged retinas following treatment with the *foxj1a* morphant. However, while the *pcna* and *foxj1a* morphants were equally effective at inhibiting proliferation in the NMDA-treated retina (Fig. 7d, e), the *foxj1a* morphant was less effective than the *pcna* morphant at doing so in light-damaged retinas (Fig. 7g, h). Furthermore, at 36 h following injury, while the *pcna* morphant was effective at inhibiting

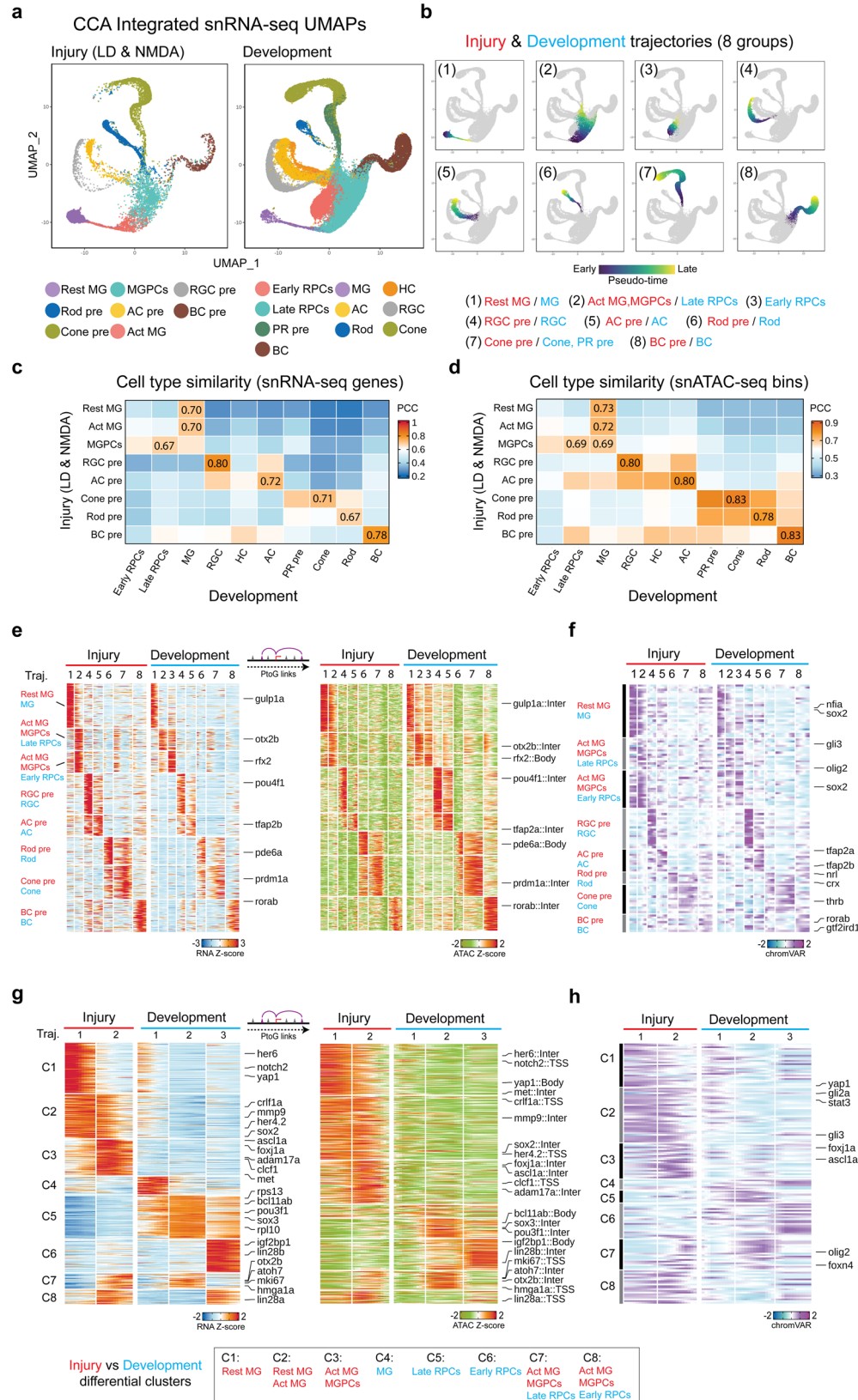

**a** CCA Integrated snRNA-seq UMAPs

**b** Injury & Development trajectories (8 groups)

(1) Rest MG / MG  (2) Act MG,MGPCs / Late RPCs  (3) Early RPCs
(4) RGC pre / RGC  (5) AC pre / AC  (6) Rod pre / Rod
(7) Cone pre / Cone, PR pre  (8) BC pre / BC

**c** Cell type similarity (snRNA-seq genes)

**d** Cell type similarity (snATAC-seq bins)

**e**

**f**

**g**

**h**

Injury vs Development differential clusters

C1: Rest MG
C2: Rest MG Act MG
C3: Act MG MGPCs
C4: MG
C5: Late RPCs
C6: Early RPCs
C7: Act MG MGPCs Late RPCs
C8: Act MG MGPCs Early RPCs

proliferation of INL cells (which correspond to proliferating MG at this stage) in both NMDA-treated and light-damaged retina, the *foxj1a* morpholino only reduced proliferation in NMDA-treated retinas at this stage, and had no significant effect in light-damaged retina (Fig. S8a–f). At this time, proliferating ONL cells represent rod precursors[46], which are not derived from injury-induced MGPCs and do not take up either

morpholino, and as expected showed no change in either morphant (Fig. S8a–f). Loss of function of *foxj1a* therefore preferentially affected MG proliferation in NMDA-treated retina relative to light-damaged retina. All cell counts are listed in Source Data.

To determine if *foxj1a* is required for developmental retinal neurogenesis, wild-type embryos were injected at the 1–4 cell stage with

**Fig. 5 | Shared and differential features of MG-derived cells between injury and development datasets. a** Integrated UMAP projection of Müller glia (MG) and progenitor neuron cells using injury (light damage (LD) and NMDA; left) and development (right) snRNA-Seq datasets. Each point (cell) is colored by cell type in each dataset. Resting (Rest); Activated (Act); MG-derived progenitor cells (MGPCs); amacrine cell (AC); retinal ganglion cell (RGC); bipolar cell (BC); precursor (pre); retinal progenitor cell (RPC); photoreceptor (PR); horizontal cell (HC). **b** UMAPs showing the eight trajectories (groups) constructed from the integrated UMAPs of combined injury and development datasets. Color indicates pseudotime state. The label indicates the cell types included for each trajectory. The heatmap displays the

Pearson correlations between the cell types from the injury and development datasets using snRNA-Seq RNA expression (**c**) and snATAC-seq bin signals (**d**). The highest correlation score for each injury cell type is labeled on the heatmap. **e** Heatmap shows the consensus marker genes and their related marker peaks (TSS and enhancer) between injury and development model for each group. **f** Heatmap shows the consensus motifs between injury and development model for each group. **g** Heatmap shows the differential genes and their related differential peaks (TSS and enhancer) between injury and development model for MG groups (group1 and group2). **h** Heatmap shows the differential motifs between injury and development model for MG groups (group1 and group2).

the Standard Control morpholino, *mmp9* morpholino, or the *foxj1a* morpholino. At 96 hpf, all the embryos appeared normal except for the *foxj1a* morphants, which possessed significantly curled body axes (Fig. S8g), which is consistent with the previously published *foxj1a* mutant phenotype[47]. At 96 hpf, we examined retinal organization by immunostaining (Fig. S8g). Both the uninjected and the Standard Control morphant retinas possessed normal lamination and wild-type staining for rod photoreceptors, HuC/D (amacrine and ganglion cells), and GFAP (MG; Fig. S8g). While the *foxj1a* morphant retinas possessed normal lamination and all the antibodies stained the expected retinal layers, the overall maturation of the retina appeared somewhat delayed relative to the control retinas (Fig. S8g). It is unclear if this delayed retinal development seen in *foxj1a* morphants is due to its previously documented widespread early developmental function[47], or is mediated by disrupted function of the very small number of *foxj1a*-expressing late-stage retinal progenitor cells. These findings demonstrate a selective role for Foxj1a in regulating injury-induced neurogenesis in zebrafish retina.

## Discussion

In this study, we demonstrate that different retinal damage models induce a similar cellular response by which the MG reprogram and divide asymmetrically to produce MG-derived progenitor cells that continue to proliferate and differentiate into retinal neurons (Fig. 8). However, there are distinct differences in the gene expression profiles and TF networks between these two retinal damage models. For example, *mmp9*, which is expressed at a higher level in NMDA-damaged retinas relative to light-damaged retinas, selectively regulates generation of set of MGPC-derived retinal cell types (amacrine and ganglion cell cells) relative to another (photoreceptors). Furthermore, while these two damage models produce multipotent MGPCs that are transcriptionally similar to late retinal progenitors, there are several differences in both gene expression and transcriptional regulatory networks between MGPCs and bona fide late-stage progenitors from developing retina. This is exemplified by the identification of specific genes, such as *foxj1a*, that significantly repress retinal regeneration while only modestly delaying retinal development. Taken together, these results reveal that while there are similarities in the gene regulatory networks that control MG reprogramming in different damage models and between regenerating and developing retinas, key differences exist that highlight the finding that injury-induced regeneration does not directly recapitulate developmental neurogenesis, and that different gene and TF networks are required for the proper regeneration of retinal neurons in different damage environments.

Our comparative analysis of MGPCs following light damage and NMDA injury reveals several common and often unexpected features. We observe that MGPCs mostly produce rod and cone photoreceptors, amacrine cells, and RGCs regardless of the injury model, while generating few bipolar and no horizontal cells. MGPCs also show inherent biases in neurogenic competence that are determined by the specific neuronal cell types lost to injury. Light damage preferentially induces generation of rods and cones, while NMDA preferentially generates RGCs. However, there is extensive generation of most retinal neuronal

cell types from MGPCs in both injury models, and there appears to be minimal postmitotic elimination of MGPC-derived neurons.

These biases in neurogenic competence are likely the result of differential transcriptional response to extrinsic signals. These could potentially include signals released directly from dying neurons, the loss of contact-mediated or secreted signals from these cells, secreted factors selectively induced by injured neurons, and signals derived indirectly from other cell types such as microglia[48,49]. Light damage induces substantially higher activity of neurogenic bHLH factor *ascl1a*, while NMDA injury triggers a more typical inflammatory response, marked by particularly high expression and activation of *stat1/2* genes. Notably, selective *stat1/2* induction is also observed in microglia following NMDA injury, as well as induction of *interleukin 1-beta* (*il1b*) and other cytokines (Fig. 4g), and this may in turn induce changes in gene expression in MG.

MGPCs show substantially higher expression of several TFs that promote formation and/or survival of RGCs, such as *atoh7*, and these likely play an important role in restricting developmental competence. While light damage induces higher levels of expression of *foxj1a* relative to NMDA-treated retinas (Fig. 2g), *foxj1a* morphants show a less dramatic reduction in MGPC proliferation in light-damaged retinas (Fig. 7). This may result from there simply being more functional Foxj1a protein present in light-damaged retinas following morpholino treatment, or else may point to unexpected regulatory functions of this transcription factor, which await further functional investigation. Although, like virtually all cells, Müller glia possess primary cilia[50], *foxj1a* is well-characterized as a selective master regulator of the development of multiciliated ependymal cells[51], which are not present in either the developing or regenerating retina. No evidence for selective expression of genes specific to motile cilia, or other known markers of multiciliated cells, is seen in the regenerating retina (Supplementary Dataset 5). The recent finding that Foxj1 regulates mammalian cortical cell fate specification independent of any role in ciliogenesis[52] identifies new potential mechanisms of action in the context of MGPC proliferation.

Secreted factors such as Mmp9 likewise play an important role in selectively controlling generation of specific cell types. Mmp9 has been previously shown to inhibit generation of MGPC-derived photoreceptor precursors following light damage[36]. While we replicate this observation here, we also observe a stronger and more selective effect on inhibiting generation of amacrine and ganglion cells following both light damage and NMDA injury. The fact that *mmp9* expression is enriched in MGPCs in the NMDA-treated retina, which are biased toward generating ganglion cells, underscores the importance of precise control of injury-induced neurogenesis through active negative regulation. The precise mechanism of action of Mmp9 in this context remains unclear. Mmp9 is a secreted protease that cleaves components of the extracellular matrix such as collagen and laminin[53], as well as multiple chemokines[54] and lipid binding proteins[55], and both MG and microglia differentially express multiple genes in all these functional categories following NMDA treatment and light damage (Supplementary Dataset 4: T5 and T8). Most notably, Mmp9 also controls maturation of Il1b[56], which is selectively upregulated in

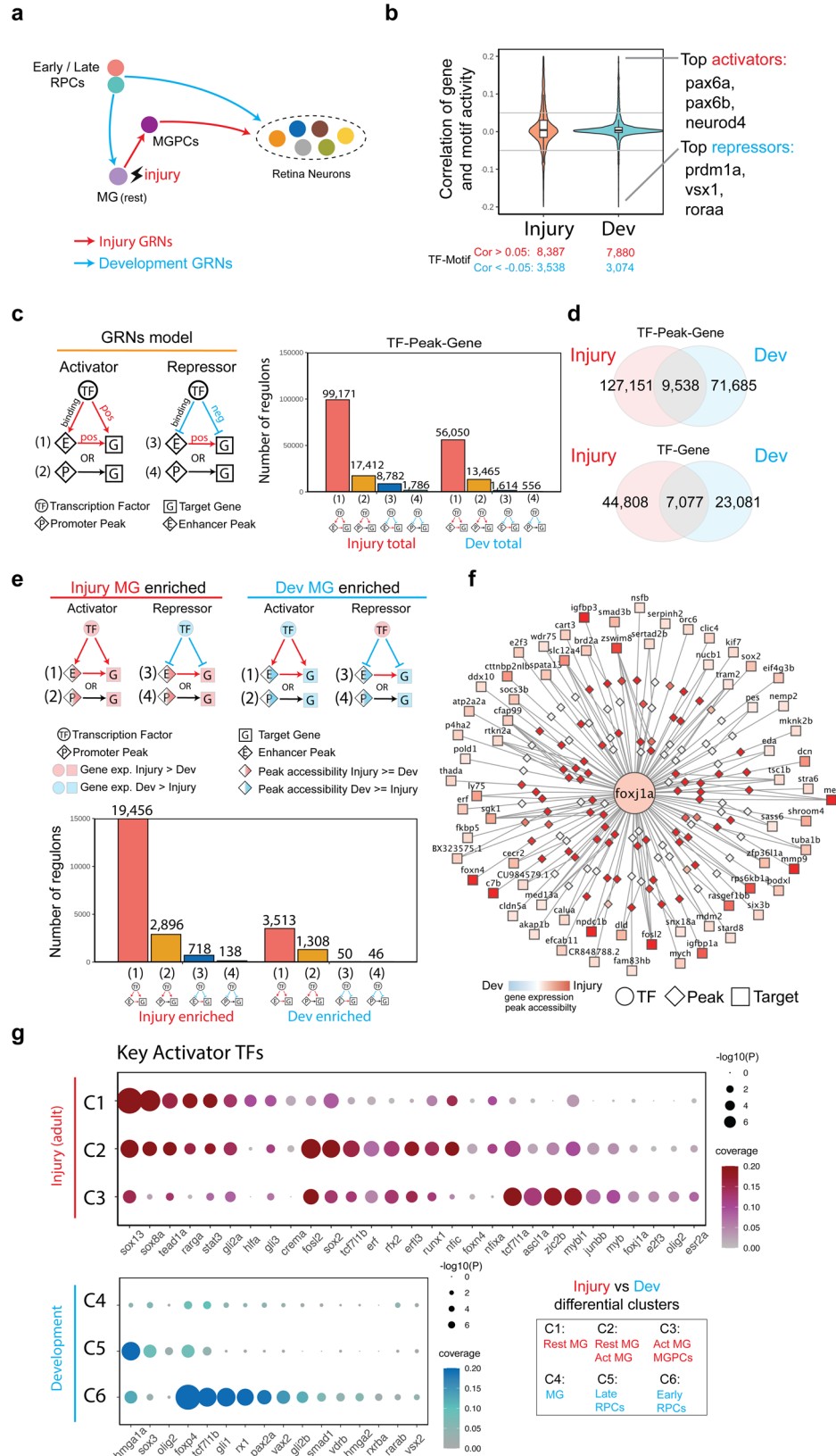

**Nature Communications** | (2023) 14:8477

microglia following NMDA injury (Fig. 4g). Further experiments are required to determine the mechanism by which Mmp9 selectively inhibits generation of inner retinal neurons.

It was previously described that zebrafish retinal development begins with the commitment of retinal ganglion cells, followed by amacrine cells, then the cone photoreceptors and ultimately bipolar cells, rod photoreceptors and Müller glia[42]. Our snRNA-Seq data is consistent with this temporal pattern with some additional refinement (Supplementary Dataset 5). For the INL cells, amacrine cells appear to be generated immediately following the ganglion cells. The bipolar cells and horizontal cells are next and, as described previously[42], we then observe generation of cone photoreceptors, followed by the rod

**Fig. 6 | Transcription factors controlling differential expression genes in MGPC in injured retina and progenitor cells in developing retina. a** Schematic of gene regulatory networks (GRNs) that direct retinal progenitor cell (RPC) differentiation during development, Müller glia (MG) reprogramming following injury, and MG-derived progenitor cell (MGPC) differentiation during regeneration. **b** Inference of activator and repressor function for each individual transcription factor from multiomic datasets. The *y*-axis represents the correlation distribution between gene expression and chromVAR score. The top three activator and repressor TFs are shown on the right. The center, lower/upper bound of the boxplot shows the median, 25th and 75th of the correlations, The whiskers extend from the ends of the box to the smallest and largest values that are within 1.5 times the IQR from the lower and upper quartiles, respectively. **c** Gene regulatory networks of injury and development datasets. (left) Triple regulons model, A circle indicates a TF, a rectangle indicates a target gene, and a diamond indicates a peak (right) barplot shows

the number of types of regulons. **d** Venn diagram shows the overlap of regulons between injury and development GRNs. **e** Enriched gene regulatory networks of injury and development. (Left) Enriched Triple regulons model for each condition. A circle indicates a TF, a rectangle indicates a target gene, and a diamond indicates a peak. Color indicates the log2 fold change of gene/peak between injury and developmental datasets. (Right) barplot shows the number of different types of regulons between injury and development enriched GRNs. **f** An example of *foxj1a* regulons. Color indicates the log2 fold change of gene/peak between injury and developmental datasets. A circle indicates a TF, a rectangle indicates a target gene, and a diamond indicates a peak. **g** Dotplot showing key activator TFs for each divergent gene cluster. The color of the dot showing the gene ratio and the size indicates the *p* value of the TF's regulatory specificity of the regulatory relationship. Hypergeometric test were used test whether the TF's targets are enriched in the DEG clusters.

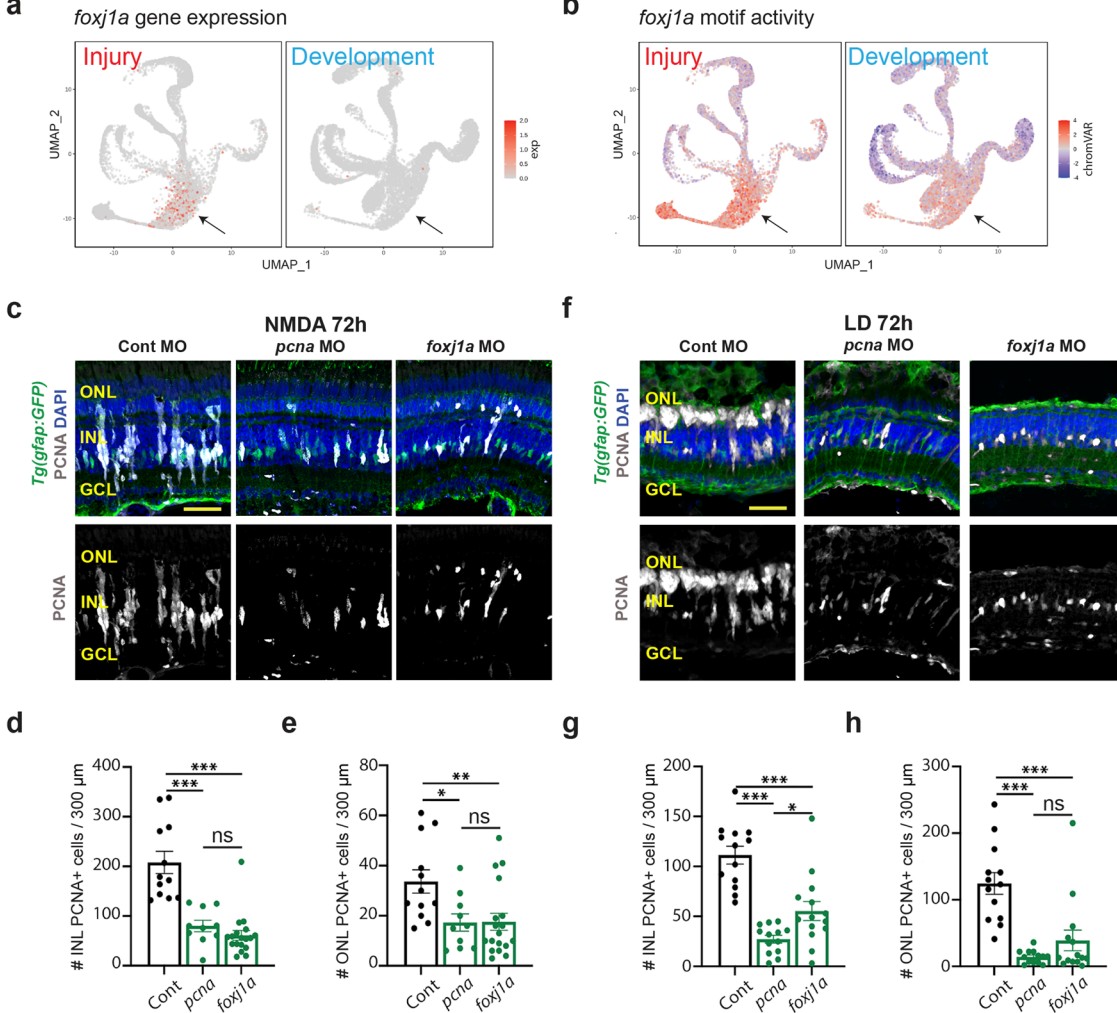

**Fig. 7 | *foxj1a* is required for MGPC proliferation. a** UMAPs showing the gene expression pattern of *foxj1a* between injury (left) and development model (right). **b** UMAPs showing the chromVAR motif activity of *foxj1a* between injury (left) and development model (right). **c** Tg(*gfap:GFP*) retinas electroporated with either Standard Control morpholino (Cont MO), *pcna* MO, or *foxj1a* MO were isolated 72 h after NMDA injection and immunostained for PCNA, GFP, and counterstained with DAPI. Outer nuclear layer (ONL); inner nuclear layer (INL); ganglion cell layer (GCL). **d** Quantification of the number of PCNA-labeled cells in the INL. Cont *n* = 12, *pcna* *n* = 10, *foxj1a* *n* = 18. Three independent experiments. **e** Quantification of the number of PCNA-labeled cells in the ONL. Cont *n* = 12, *pcna* *n* = 10, *foxj1a* *n* = 18. Three independent experiments. **f** Tg(*gfap:GFP*) retinas electroporated with either

Cont MO, *pcna* MO, or *foxj1a* MO were isolated after 72 h of light damage (LD) and immunostained for PCNA, GFP, and DAPI. **g** Quantification of the number of PCNA-labeled cells in the INL. Cont and *pcna* *n* = 13, *foxj1a* *n* = 14. Three independent experiments. **h** Quantification of the number of PCNA-labeled cells in the ONL. Cont and *pcna* *n* = 13, *foxj1a* *n* = 14. Three independent experiments. Scale bars in (**c**) and (**f**) are 20 μm. **d**, **e**, **g**, **h** Data are presented as mean values ± SEM. For statistical analysis, one-way ANOVA was followed by *t*-test with Dunnett's method for multiple comparisons correction. Asterisks indicate statistically significant differences between the indicated groups (**p* ≤ 0.05, ***p* ≤ 0.01, ****p* ≤ 0.001). Source data are provided as a Source Data file 1.

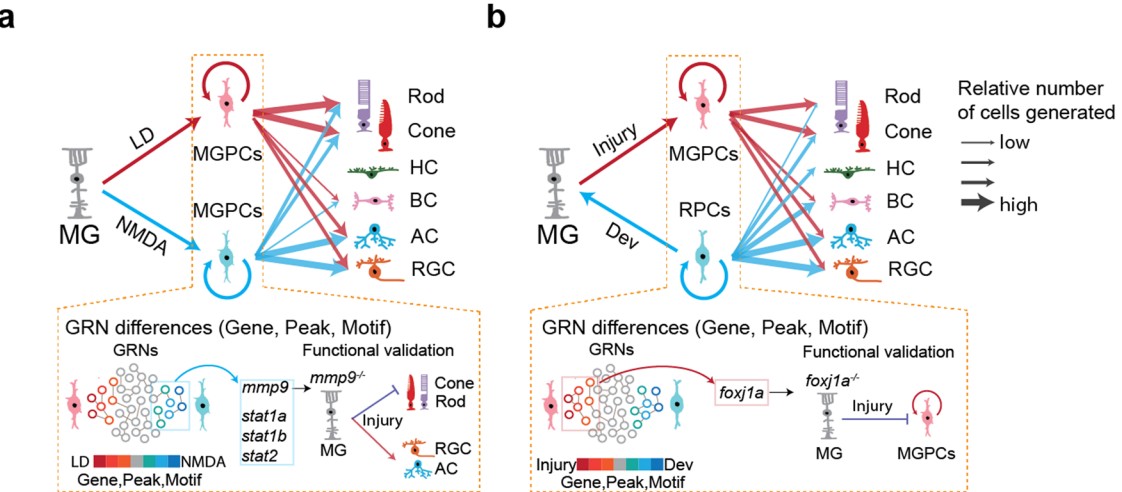

**Fig. 8 | Schematic summary of key findings from this study. a** Summary of major differences between MGPCs induced by light damage and NMDA excitotoxicity. **b** Summary of major differences between MGPCs and RPCs induced by injury and during development.

photoreceptors. Transcriptionally distinct MG precursors are generated between the cone and rod photoreceptors (Fig. S4a). This suggests a model of regeneration where the MG reprogram and produce MGPCs that are very similar to late-stage RPCs, which first generate ganglion cells, followed by INL neurons, and then cone and rod photoreceptors. We propose that this results in the differentiation of all retinal neuronal classes, regardless of the neuronal classes lost due to damage. However, if the early differentiating neuronal classes experienced little, or no, cell loss, then the majority of the MGPCs are retained and differentiate into photoreceptors. In contrast, if the damage results in primarily inner retinal neuronal classes being lost, as is the case with NMDA excitotoxicity, then the majority of the MGPCs must differentiate into these lost neuronal cell types and only a subset of the MGPCs are retained to differentiate into cone and rod photoreceptors. This model is in addition to the damage-specific transcriptional changes that we observed, which can further refine the commitment of the MGPCs into the differentiated neuronal ratios that were observed.

While MGPCs closely resemble RPCs at the molecular level, a number of major differences are evident that likely correspond to known differences in proliferative and/or developmental competence. RPCs show a distinct early-stage state that is not observed in MGPCs. This does not appear to strictly correspond to an early state of developmental competence, as has been reported in mammals[57–59], since early-born cell types such as RGCs, horizontal cells and cone photoreceptors are still being generated at 48 hpf, when RPCs have adopted a late-stage profile. These differences between RPCs and MGPCs may instead reflect differences in relative levels or patterns of proliferation, although this possibility remains to be investigated. Furthermore, while RGCs are efficiently generated by MGPCs, they are generated at much higher relative levels during early stages of developmental neurogenesis, which likely reflects that fact that a large majority will be eventually eliminated by apoptotic cell death[60]. The relatively weaker expression of RGC-promoting factors such as *sox11a* in MGPCs may mediate these differences in RGC generation, while the absence of horizontal cell-promoting TFs such as *onecut1* and *pou2f2a* likely underpins the lack of competence to generate horizontal cells seen in MGPCs in these injury models. It remains to be determined whether selective conditional ablation of either cell type[61] might activate expression of these genes in MGPCs. Furthermore, since robust methods of inducing MG-derived neurogenesis have now been developed in mammals[41,62–64], these findings raise the possibility that different modes of injury might also differentially regulate generation of specific MG-derived neuronal subtypes in this context.

## Methods

### Ethics statement
All zebrafish experiments were performed in accordance with the regulations of University of Notre Dame (A3093-01). All the experiment protocols were approved by the University of Notre Dame (21-02-6420 and 21-02-6437) and Johns Hopkins University School of Medicine.

### Zebrafish husbandry
Adult *albino*[b4/b4 65,66], *mmp9*[mi5003] mutant fish[36], and transgenic *albino Tg(gfap:EGFP)*[ntlI19] zebrafish (*Danio rerio*, AB strain) used in this study were bred and maintained in the Freimann Life Science Center Zebrafish Facility at the University of Notre Dame in accordance with standard operating policies and procedures[2,36]. The fish were kept at a temperature of ~26.5 °C and 30% humidity, with a 14-h light, 10-h dark cycle[67]. *Tg(gfap:EGFP)*[ntlI] was used to label the MG by expressing GFP[19]. All the zebrafish used in this study were adults between 6 and 12 months old or embryos. Prior to any experiments, the fish were anesthetized or euthanized using 1:1000 or 1:500 2-phenoxyethanol (Sigma), respectively. Animals of both sexes were used in approximately equal numbers for these studies.

### Retinal damage and EdU labeling
Phototoxic treatment that causes photoreceptor ablation was used to initiate a regenerative response in the zebrafish retina[2,68]. All fish subjected to light damage were dark-adapted for 2 weeks and then placed adjacent to intense fluorescent lights for 96 h. The temperature of the fish tanks was regularly monitored and maintained at 34 °C. After 96 h, the fish were returned to the animal facility with a normal light/dark cycle and maintained for 7, 14, or 21 days recovery (DR).

N-methyl-D-aspartate (NMDA, Sigma M3262) was administered to normal light/dark cycled fish via an intravitreal injection of 0.3 µl 100 mM NMDA solution in sterile PBS with a Hamilton syringe (Hamilton, 2.5 µl Model 62 RN SYR, 87942), followed by placing the tanks in a dark incubator at 33 °C for 96 h, after which the tanks were moved back into a normal light/dark cycle for 7 or 14 days recovery.

EdU (5-Ethynyl-2′-deoxyuridine, Life technologies A10044) was administered intraperitoneal via[68]. Fish were anesthetized using 1:1000 2-phenoxyethanol and immobilized under a paper towel in a petri dish. Then, using an insulin syringe equipped with a 30-gauge needle, ~20 µl

1 mg/ml EdU in water was injected intraperitoneally into each zebrafish at 60, 72, 84, 96 and 108 h after damage. The retinal tissues were harvested at either 3, 7, 14, or 21 days recovery.

## In vivo morpholino-mediated gene knockdown

Conditional adult retinal morphants were generated essentially as described previously[44,45]. Prior to light treatment, fish were anesthetized in 1:1000 2-phenoxyethanol, the outer cornea membrane of the left-eye was removed with Dumont #5 forceps, an incision was then made though the inner cornea in the ventral-caudal aspect of the pupillary margin using a #11 scalpel, and 0.2–0.3 μl of morpholinos (Gene Tools, LLC) at a concentration of 1 mM were injected into the posterior segment using a 2.5 μl Hamilton syringe. Morpholinos that do not target any known zebrafish genes (Standard Control), as well as one targeting *pcna*, which blocks MG proliferation, were used as negative and positive controls, respectively. Morpholinos were electroporated into the retina using platinum electrode tweezers with a CUY21EDIT Square Wave Electroporator (Nepa Gene Company Ltd.), delivering two 50 msec pulses, separated by a 1 s pause, at 0.75 Volts, resulting in the current of 0.05 Amperes. The electrodes were used to prolapse the eye out of the socket, with the anode in contact with the cornea, and the cathode positioned 2 mm behind the dorsal part of the eye. Successful electroporation was confirmed by the presence of lissamine fluorescence within the retinal cells using microscopy. For the embryos conditional morpholinos mediated knock-down experiments, the morpholinos were diluted to 1 mg/ml in 1x Danieau buffer. Wild-type or *Tg(gfap:EGFP)*[nt11] embryos were injected with 5 ng of morpholinos at the one-cell stage[69].

Morpholino sequences used were:

Standard Control: 5′ CCTCTTACCTCAGTTACAATTTATA 3′ (Gene Tools, LLC), anti-*pcna:* 5′ TGAACCAGACGTGCCTCAAACATTG 3′[45], anti-*foxj1a:* 5′ CATGGAACTCATGGAGAGCATGGTC 3′[70], and anti-*mmp9:* 5′ ACTCCAAGTCTCATTTTGAGTCGCA 3′ (Gene Tools, LLC).

## Tissue fixation and cryosectioning

Tissues for immunocytochemical analyses were collected at specified time points by euthanizing zebrafish in 1:500 solution of 2-phenoxyethanol, followed by ocular enucleation using #5/45 Dumont forceps (Supplier). The eyes were floated in 1x Phosphate-Buffered Saline (PBS) solution, and a circular opening, approximately the size of the pupil, was made in the cornea using micro-scissors, creating an eye cup. The eye cups were fixed at room temperature (RT) in 4% paraformaldehyde, using a pH shift technique to preserve the cytoskeletal structure and avoid the cellular swelling that is typically associated with standard paraformaldehyde fixation at 4 °C[71]. Briefly, the tissues were fixed at RT for 5 min in 80 mM HEPES, 2 mM EGTA, 2 mM $MgCl_2 \cdot 6H_2O$, 4% PFA, pH 6.8–7.2, followed by 30 min in 100 mM $Na_2B_4O_7 \cdot 10H_2O$, 1 mM $MgCl_2 \cdot 6H_2O$, 4% PFA, pH 11. The eyes were washed in 1x PBS at RT and dehydrated in a 15% sucrose solution at 4 °C overnight. The eyes were then placed in a 2:1 solution of PolyFreeze Tissue Freezing Medium (TFM) (Sigma-Aldrich) and 30% sucrose for 24 h at 4 °C. Finally, the eyes were arranged in the dorsal-ventral plane in 100% TFM, frozen, and stored at −80 °C. Frozen tissue blocks were sectioned using a Thermo Scientific Microm HM550 Cryostat at −23 °C, producing 14 μM serial sections that included the optic nerve and both the dorsal and ventral crescent. Retinal sections were collected on SuperFrost glass slides (VWR 48311-703), dried for 30 min on a slide warmer at 50 °C, and stored at −80 °C for future immunohistochemical staining.

## Immunohistochemistry

Retinal sections were processed for immunohistochemistry by drying the slides at 50 °C for 20 min and followed for 3 washes in 1x PBS[2,68]. Sections were blocked for 1 h in blocking buffer (1x PBS, 4% normal goat serum, 0.4%Triton X-100, 2% DMSO). When anti-PCNA primary antibodies were used, an extra heat-induced antigen retrieval step was performed. Prior to blocking, slides were immersed in a 10 mM sodium citrate, 0.05% Tween 20, pH 6 solution, and heated for 25 s in a microwave at maximum power to nearly reach the boiling point. The slides were heated for another 7 min using the microwave at 10% power in periodic 20–25 s bursts to maintain near-boiling temperature. The slides were cooled for 30 min at RT and washed 2 × 5 min in 1x PBS before proceeding to the blocking step.

The following primary antibodies were used: mouse anti-PCNA monoclonal antibody (Sigma P8825, 1:500 dilution), rabbit anti-GFAP (Dako Z0334, 1:300 dilution), mouse anti-HuC/D monoclonal antibody (Invitrogen A21271, 1:300 dilution), mouse anti-4c12 monoclonal antibody (gift from Dr. Fadool, Florida State University, 1:200 dilution), rabbit anti-PKCa (Sigma Life Science P4334, 1:300 dilution), rabbit anti-UV cone opsin[72], 1:1000 dilution), rabbit anti-green cone opsin[72], 1:500 dilution), mouse anti-Zpr1 monoclonal antibody (ZIRC, 1:200) rabbit anti-Lcp1 (GeneTex GTX134697, 1:500). Secondary antibodies (diluted 1:500) included: goat anti-mouse 488 (Life Technologies A11029), goat anti-mouse 594 (Life Technologies A11032), goat anti-mouse 647 (Life Technologies A21236), goat anti-rabbit 488 (Life Technologies A11034), goat anti-rabbit 594 (Life Technologies A11037), goat anti-chicken 488 (Life Technologies A11039).

## Confocal microscopy and quantification

A Nikon A1R HD inverted confocal microscope at the University of Notre Dame Integrated Imaging Facility was used to capture serial optical sections of retinal slices using 40x and 60x oil-immersion objectives. Z-stacks of ~10 μm from the central dorsal retinal tissue were captured using NIS-Elements (RRID:SCR_014329). The resulting images were analyzed as either single optical sections or maximum z-stack projections (where appropriate) using ImageJ/FIJI[73]. For the colocalization experiments, single optical sections were used. We increased the saturation of the Gfap and 4c12 channels to increase the cell body staining to determine colocalization with EdU.

## Statistical analysis

The resulting data were analyzed using GraphPad Prism version 9. Student's *t* test for single pairwise comparisons with control or one-way or two-way analysis of variance followed by Dunnett's or Bonferroni's post hoc test for multiple comparisons. The statistical significance is indicated in each graph as follows: * for $p \leq 0.05$, ** for $p \leq 0.01$, *** for $p \leq 0.001$, or ns for not significant difference.

## ScRNA-Seq of methanol-fixed retinal cells

Zebrafish retinas (4–6 retinas from 4–6 fish) were collected at different time points after NMDA and LD treatments. Retinal cells were dissociated and fixed in −20 °C methanol and stored in −80 °C freezer until used. Fixed cells were recovered and subjected to 10x Genomics Single-Cell system as described previously[41]. Libraries were prepared and sequenced on Illumina NovaSeq at ~500 million reads per library.

## Multiome analysis of frozen retinal tissues

Zebrafish retinas (4–6 retinas from 4–6 fish) were collected at different time points after NMDA and LD treatments, and flash-frozen in dry ice for ~15 min before being transferred to a −80 °C freezer for storage. Nuclei were extracted from frozen retinal tissues according to 10xMultiome ATAC + Gene Expression (GEX) protocol (CG000338). Briefly, frozen retinal tissues were lysed in ice-cold 500 ml of 0.1X Lysis buffer using a pestle and incubated on ice for 6 min totally. Nuclei were centrifuged, washed 3 times and resuspended in 10xMultiome nuclei buffer at a concentration of ~3000–5000 nuclei/ml. Nuclei (~15k) then were loaded onto 10x Genomic Chromium Controller, with a target number of ~10 K nuclei per sample. RNA and ATAC libraries were prepared according to the 10xMultiome ATAC + Gene Expression protocol and subjected for Illumina NovaSeq sequencing at ~500 million reads per library.

## Single-cell muti-omics analysis

**Preprocessing.** The multi-omic sequencing files were processed for demultiplexing and analyzed using Cell Ranger ARC v2.0. The genes were mapped and referenced using the zebrafish reference genome DanRer11. To address potential ambient RNA contamination in the cell-by-gene expression matrix, we employed scAR[74] module from scvi-tools[75] with default parameters for ambient RNA cleaning for each sample.

**Filtering barcode doublets and low-quality cells for each sample.** For snRNA-seq injury analysis, the cell-by-genes matrices were converted to Seurat v3 objects[76]. Doublet cells were identified using the solo module[77] of scvi-tools. Cells were filtered out if their Solo doublet score were greater than zero. Furthermore, cells with low RNA counts (nCount_RNA < 500) as well as cells with high RNA counts (nCount_RNA > 50,000) were also filtered out as low-quality cells. For snATAC-seq analysis, the fragment files were initially converted to ArchR[78] objects with default parameters. We only kept the cells which passed quality control in the scRNAseq datasets. Additionally, the cells were further filtered out if their doublet enrichment score were greater than 2. For the development multi-omics datasets, we used the same methods to filter out low-quality cells. However, we only kept cells that passed quality control in both the snATAC-seq and scRNA-seq.

**Clustering, visualization, and identification of cell types.** The clustering analysis was conducted on the combined dimensional space obtained from both the RNA and ATAC single-cell matrices. Initially, the RNA expression matrix was incorporated into the ArchR project using the "addGeneExpressionMatrix" function. Subsequently, dimension reduction was separately applied to the cell-by-bin matrix and the cell-by-gene matrix within the ArchR project using the "addIterativeLSI" function. The resulting reduced dimensions were then merged into a new dimension called "LSI_Combined" using the "addCombinedDims" function. Finally, clustering was performed using the "addClusters" function with a resolution of 1.5, taking into account the "LSI_Combined" dimensions. To visualize single cells, the UMAP embeddings were calculated using the "addUMAP" function in ArchR with the first 10 "LSI_Combined" dimensions.

To infer cell types, the existing zebrafish scRNA-seq data[41] were utilized for interpreting our snRNA-seq cell types using the CCA (canonical correlation analysis) integration method in the Seurat package[79]. Firstly, the zebrafish scRNA-seq data were downloaded and converted into Seurat objects. Secondly, for each snRNA-seq sample, the anchors between them were identified using the "transfer.anchors" function, and the "TransferData" function was used to obtain the cell type prediction results for each cell. Cells with a prediction score <0.5 were further filtered out, and each cluster was annotated based on their predicted cell types. To further identify precursor cell clusters, the clusters in each major cell type between the control and injury datasets were compared. Clusters that existed in the injury dataset but not in the control dataset were selected as precursor cell clusters and were further confirmed using specific precursor cell markers.

**Generating fixed-width and non-overlapping peaks.** Fixed-width 501 bp peaks were called for each cell type in both the injury and development datasets by MACS2[80] in ArchR package. Subsequently, these peaks were merged using an iterative method to retain the most significant peak among overlapping peak sets. The function "addReproduciblePeakSet" was utilized for this calling and merging process. Following that, the peak matrix was generated for each dataset using the function "addPeakMatrix" with the following parameters: ceiling = 4 and binarize = FALSE.

## Single-cell RNA-seq data analysis

**Preprocessing.** The scRNA sequencing files were processed for demultiplexing and analyzed using Cell Ranger. The genes were mapped and referenced using the zebrafish reference genome DanRer11. To address potential ambient RNA contamination in the scRNA-seq expression matrix, we employed SoupX[81] with default parameters for further ambient RNA cleaning for each sample.

**Quality control of scRNA-Seq data for each sample.** The cell-by-genes count matrices were converted to Seurat v3 objects[76]. Cells with RNA counts less than 500 or greater than 50,000 were filtered out as low-quality cells. Additionally, cells with a mitochondrial fraction of greater than 15% were also removed. Doublet cells were identified using the Scrublet Python package[82].

**Clustering, visualization, and cell type annotation of scRNA-Seq data.** The control, LD and NMDA samples were first integrated using the Seurat integration functions (SelectIntegrationFeatures, FindIntegrationAnchors and IntegrateData). Data scaling, dimensional reduction and clustering were then performed on the integrated injury dataset using the standard Seurat pipeline. For visualization, combined UMAP were generated using the first 20 dimensions. We used the same methods to infer cell types as we did for single-cell multiomics data. Briefly, the existing zebrafish scRNA-seq data[41]) were utilized for interpreting our scRNA-seq cell types using the CCA (canonical correlation analysis) integration method in the Seurat package.

**Integration of injury and development single-cell RNA-seq data.** The similarity between the injury and development datasets was assessed by integrating them based on their single-cell expression profiles using the reciprocal PCA workflow from Seurat. The integration process was performed with a strength parameter of k.anchor = 5. Subsequently, integrated UMAPs were generated using the first 1 to 15 dimensions derived from the integrated dimensions. The cell type annotations on the new integrated dataset remained the same as those in the separate injury and development datasets.

## Trajectory inference and pseudotime analysis

Slingshot software[83] was utilized to infer pseudotime based on the UMAP coordinates, which were filtered by selecting cells involved in the specific biological process under investigation.

For the injury datasets, the LD and NMDA combined UMAP coordinates were first separated based on the cells from the 6 groups: MG (Rest MG, Activated MG and MGPCs); RGC pre; AC pre; BC pre; Cone pre; and Rod pre. Then, Slingshot was applied to infer trajectories using the "getLineages" and "getCurves" functions. In the Rest MG to MGPCs group, the cell cluster overlapping with "Rest MG" was treated as the "root cluster". For the other groups, the closest cell cluster to "MGPCs" was treated as the "root cluster".

For the combined injury and development datasets, Slingshot was run to construct MG-related trajectories using the merged scRNA-seq UMAP coordinates. The cell cluster overlapping with "Rest MG" in the injury dataset and "MG" in the development dataset was treated as the "root cluster" for Slingshot.

Subsequently, the "slingPseudotime" function was utilized to calculate the pseudotime state for each cell. Finally, the pseudotime was merged into 20–50 bins for each trajectory, and the average gene expression, average accessibility level, and average motif activity score were calculated for each bin. The scores for each bin were further smoothed and scaled for visualization.

## Identification of differential and consensus genes, peaks, and motif activities

The consensus (CEG) and differential genes (DEG) were identified using the ambient-RNA-cleaned cell-by-gene count matrix for both

scRNA-seq and multi-omic datasets. The peaks were annotated with their predicted target genes, as described in the GRNs construction methods section. To further remove potential ambient RNA contamination from DEGs, the marker genes of mature retina neurons (Rod, Cone, RGC, AC, HC, BC, Microglia) were identified using control samples with the FindMarkers function: log2FC > 2 and adj-*p* value < 0.05. The DEG list of MG cell groups (MG, Act MG, MGPCs) in the downstream analysis excluded these mature marker genes. In the motif analysis, motifs from both cis-BP and Transfac database (2018) were collected. For each TF, the corresponding motifs were filtered based on the correlation between TF expression and motif activity (chromVAR) score, which described in the GRNs construction methods section.

To identify the DEGs of MG cell groups between the LD and NMDA condition (or between injury and development) datasets, we used combined cells from all time points within a cell type to call DEGs. We do the following comparison for LD and NDMA: LD-RestMG vs. NMDA-RestMG, LD-ActMG vs. NMDA-ActMG, and LD-MGPCs vs. NMDA-MGPCs. For injury and development, we compare: Injury-MG vs. Dev-MG, Injury-ActMG-MGPC vs. Dev-LateRPC, and Injury-ActMG-MGPC vs. Dev-EarlyRPC.

In the DEG analysis, the "findMarkers" function in the Seurat package was employed. For the LD vs. NMDA comparison, the DEGs were defined as abs(log2FC) > 0.25 and adj-*p* value < 0.05. For the injury vs. development comparison, the DEGs were defined as abs(log2FC) > 0.5 and adj-*p* value < 0.05. Finally, all the DEGs are merged, the redundant DEGs are removed, and then clustered using k-means method by their expression profile along the MG-related trajectories (Figs. 2g and 5g).

To identify the CEGs between different conditions, the "findAllMarkers" function was used to determine enriched genes for each trajectory group compared to other groups. CEGs were identified based on the following criteria: a log2FC greater than 0.5 and an adjusted *p* value less than 0.05 in both the LD and NMDA datasets (or in both injury and development datasets).

The differential accessible regions (DARs) between LD and NMDA datasets (or between injury and development) were identified using the "getMarkerFeatures" function in ArchR to measure the peak differences between LD and NMDA. DARs were identified for each cell group in the injury datasets based on the criteria of having an absolute log2 fold change greater than 0.25 and a *p* value less than 0.05. Additionally, DARs were further filtered if their predicted target genes are not DEGs.

To identify the consensus lineage-specific peaks (CARs) between LD and NMDA datasets (or between injury and development), peaks were separately identified for each group compared to other groups in each condition. CARs were considered significant if they exhibited a log2 fold change greater than 0.25 and an *p* value less than 0.05. Furthermore, CARs were excluded if they were not related with the CEGs.

The differential motifs (DMs) between the LD and NMDA treatment datasets (or between injury and development) were identified using the "FindMarkers" function from the "Signac" package. The cell-by-motif *z*-score matrix from chromVAR was converted into a Seurat object. The "LR" test method was utilized to determine the differential motif accessibility. Motifs with an absolute log2 fold change (log2FC) greater than 0.25 and an adjusted *p* value less than 0.05 were classified as DMs. Furthermore, DMs were excluded if they were found to be inconsistent with the list of differentially expressed genes (DEGs) for each cell type. In cases where multiple DMs corresponded to a DEG in the list, only the DM with the highest correlation with its corresponding DEG was retained.

The consensus motifs (CMs) between the LD and NMDA treatment (or between injury and development) were identified for each cell group by comparing chromVAR *z*-score with other groups for each condition. Motifs with an absolute log2 fold change (log2FC)

greater than 0.25 and an adj-*p* value less than 0.05 were classified as CMs. Additionally, CMs were excluded if they were inconsistent with the list of CEGs, and also the highest correlation CMs were kept in the final list.

## Constructing gene regulatory networks using muti-omics datasets

Four GRNs were constructed as follows: (1) LD GRNs, which were constructed using only the samples from LD injury datasets. (2) NMDA GRNs, which were constructed using only the samples from NMDA injury datasets. (3) Injury GRNs, which were constructed employing the samples from both LD and NMDA injury datasets. (4) Development GRNs, which were constructed using only the samples from the development datasets.

### 1. Inferring activators and repressors by expression and motif activity

For each TF-motif pair in each datasets (LD,NMDA,Injury, Development), we calculate the Spearman correlation between TF expression and motif activities (chomVAR score) at single-cell level with the "cor" function in R. Activator and repressor TF-motif pairs were identified if their correlation are large than 0.05 or less than −0.05.

### 2. Identifying cis-regulatory elements

Firstly, the categorization of all peaks into three groups based on their genomic location relative to gene loci is performed: (1) Promoter (within 500 bp of TSS), (2) Gene Body, and (3) Intergenic. Subsequently, peak-target pairs are generated using the following methods: (1) The target genes for Promoter and Gene Body peaks are determined by the genes they overlap with. (2) The target genes for Intergenic peaks encompass all genes located within 200 kb of the peak's location.

Next, PtoG correlations for each peak with its surrounding genes (200 kb) are calculated using the "addPeak2GeneLinks" function in the ArchR package. The first 30 dimensions of the combined multi-omic space are utilized to generate cell groups using ArchR.

Finally, the retention of the peak-target pairs is based on meeting the following criteria for their PtoG correlations: abs(correlation) > 0.25 and FDR < 0.01.

### 3. Predicting TF binding sites

The TF-peak pairs were constructed by predicting TF binding sites inferred based on motif information and scATACseq footprint signals within the identified cis-regulatory elements.

Initially, Position Weight Matrices (PWMs) were extracted from the TRANSFAC2018 and CIS-BP databases. The binding regions were then identified by matching these motifs to the DNA sequences of the cis-regulatory elements using the motifmatchr package[84] ("matchMotifs", p.cutoff = 5e−05).

Subsequently, the scATACseq corrected footprint signals were separately calculated for the Light-damage Injury, NMDA injury, combined injury, and Development datasets. For the Light-damage and NMDA injury datasets, the insertion fragments from Rest MG, Activated MG, and MGPCs cells were combined. For the development datasets, the insertion fragments from MG, Early RPCs, and Late RPCs cells were merged. These merged fragments were converted to BAM format and processed through the TOBIAS pipeline[85] to obtain bias-corrected Tn5 signals.

For each binding region of the motif, footprint scores were computed, including NC, NL, and NR. NC represented the average Tn5 signal at the center of the motif, while NL and NR indicated the average Tn5 signals in the left and right flanking regions (triple the size of the center) of the binding region, respectively.

Finally, the TF binding sites were retained based on the following criteria: NR + NL-2*NC > 0.1. Additionally, the binding regions for

motifs whose corresponding TFs were not expressed in the MG cells were removed.

### 4. TF-target correlation

The TF-target relationship was calculated using the Stochastic Gradient Boosting Machine (SGBM) method, which was implemented through the arboreto package[86] in Python. The "grnboost2" function generated a table of TF-gene pairs with important scores. The TF-gene pairs were filtered based on their important scores, with pairs that had scores lower than the 90th quantile being removed. Additionally, the Pearson correlation between each TF-gene pair was computed according to the cell-by-gene expression matrix. If the correlation exceeded 0.03, the TF-gene pair was annotated as "positive" regulation; if the correlation was below −0.03, it was annotated as "negative" regulation. Any other TF-gene pairs were filtered out.

### 5. Construction of TF-peak-target links

The total GRNs for each condition (LD,NMDA,Injury,and Development) were constructed by integrating data from the previous steps. The following procedure was employed: The TF-peak pairs from step 3 and the peak-target pairs from step 2 were merged to form TF-peak-target triples. Subsequently, these TF-peak-target triples were filtered using the following criteria: (1) The triples were retained only if TF activity are in the same direction with TF-gene correlation. (Activator with positive TF-gene correlation, Repressor with negative TF-gene correlation). (2) The triples were retained only if the TF's expression levels are enriched in MG cell groups (MG, ActMG, MGPCs). (3) Any duplicate triples were eliminated, and we retained the highest footprint score for each TF-peak-target pair.

### 6. Identification of enriched gene regulatory sub-networks

The enriched sub-GRNs were extracted from the total GRNs generated in step 5, based on the logFC change of TFs, peaks, and target genes (as shown in Figs. 4e and 6e). For instance, to obtain enriched LD sub-GRNs, we applied the following criteria to filter triple pairs (TF-Peak-Target) from the total LD GRNs: (1) The expression levels of TFs should be higher (or lower, if TF-motif pairs are repressors) in LD (compared to NMDA) in at least one of the MG, ActMG, and MGPCs cell groups comparisons. (2) The accessibility of peaks should not be lower in LD (compared to NMDA) in any of the MG, ActMG, and MGPCs cell groups. (3) The expression levels of target genes should be higher in LD (compared to NMDA) in at least one of the MG, ActMG, and MGPCs cell groups comparison. The same methods were employed to extract sub-GRNs from NMDA GRNs, Injury GRNs, and Development GRNs (Supplementary Dataset 4 and 7).

### 7. Identification of key activator TFs

To identify the key activators (TFs) in the GRNs, we initially remove negative regulations for each GRNs. For each TF in the network, we first calculate coverage score to see how many DEGs are regulated by that TF. For example, to identify key activators in LD enriched sub-network, for each TF, we first count the number of overlap genes between its targets with LD enriched DEG clusters (cluster 1, 2, and 3 in Fig. 4g). For each TF in each DEG cluster, the coverage were calculated as: $N_{overlap}/N_{cluster}$, where $N_{overlap}$ denote as the number of overlapped genes, and $N_{cluster}$ denote as the total number of DEGs in that cluster. Next, to test whether the given TF is specifically regulates a DEG cluster, we used hypergeometric test with "phyper" function in R for that TF and the DEG cluster. In the hypergeometric test, the "population" is defined as the TF target genes from total GRNs. The "sample" is defined as the TF target genes from LD enriched sub-GRNs, and the "successes" are defined as the genes present in both the DEGs cluster and the TF's target genes from total GRNs. Finally, all the TFs with $p$ value < 0.001 and coverage >0.01 were identified as key activators.

### ChromVAR analysis

A ChromVAR[87] analysis was performed to determine the global transcription factor (TF) activity in each cell. The raw cell-by-peak matrix was initially inputted into ChromVAR, and the "addGCBias" function was used to correct for GC bias. The DarRer11 reference genome was employed for this correction.

Subsequently, a TF $z$-score matrix was created by combining motifs from the TransFac2018 and CIS-BP databases. This was achieved using the "matchmotifs" and "computeDeviations" functions. The $z$-score of each cell was then utilized to generate a heatmap and visualization. These outputs were overlaid onto the previously calculated UMAP coordinates.

### GO analysis

We identify significantly enriched Gene Ontology (GO) terms (specifically Biological Process) and KEGG terms among different clusters of Differentially Expressed Genes (DEGs) between biological conditions, the gene set enrichment analysis was performed using the "enrichGO" and "enrichKEGG" functions from the "clusterProfiler" R package[88].

### Reporting summary

Further information on research design is available in the Nature Portfolio Reporting Summary linked to this article.

## Data availability

All the raw data for histological analysis are provided in the Source Data file. The scRNA-Seq, snRNA-Seq, and scATAC-Seq data generated in this study have been deposited in the GEO database under accession GSE239410. The re-analyzed scRNA-seq development datasets used in this study are available in the GEO database under accession code GSE135406. Source data are provided with this paper.

## Code availability

Code for GRNs construction can be found in https://github.com/Pinlyu3/Zebrafish-retina-GRNs and https://doi.org/10.6084/m9.figshare.24589434.v1.

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

## Acknowledgements

This work was supported by the Milky Way Research Foundation (to S.B., D.R.H., and J.Q.), and by NIH grants, R01EY007060, R21EY034182, P30EY007003, R01EY034493 and an unrestricted grant from the Research to Prevent Blindness (to P.F.H.). We thank Jeremy Nathans, Alex Kolodkin, Jeff Mumm, and Andrew Fischer for comments on the manuscript.

## Author contributions

S.B., J.Q., and D.R.H. conceived and supervised the study. P.L. analyzed all the mutiomic data generated, while Y.Z. analyzed all the scRNA-Seq data. M.I. and D.S. generated and analyzed all of the NMDA-treated and light-damaged data, and conducted all functional studies of mmp9 and foxj1a in adult retina, with assistance from P.B. T.H. generated all the scRNA-Seq, snRNA-Seq, and scATAC-Seq data, with assistance from I.P. L.J.C. conducted all analyses of foxj1a morphants in the early embryo. M.N., N.J.S., and P.F.H. generated and provided the mmp9 mutants. P.L., M.I., D.S., Y.Z., J.Q., D.R.H., and S.B. drafted the manuscript. All authors edited the manuscript.

## Competing interests

S.B. co-founded, and is a shareholder of, CDI Labs LLC. The remaining authors declare no competing interests.
