## [Peer Review File · Nature Communications]

Common and divergent gene regulatory networks control injury-induced and developmental neurogenesis in zebrafish retina.REVIEWER COMMENTS

Reviewer #1 (Remarks to the Author):

Müller glia in the zebrafish retina respond to injury by generating all retinal cell types. Different injury paradigms lead to the preferential loss of different retinal cell types. This offers the unique possibility to assess whether the reaction of Müller glia also preferentially replaces the lost cell type(s) over the others and which gene expression/regulatory elements are responsible for any differences. Furthermore, they address whether regenerative neurogenesis recapitulates developmental neurogenesis and find it doesn't. The authors present a thorough analysis of the regulation of gene expression and regulatory elements and discover a number of interesting functional interactions. Most importantly, *mmp9* is shown to regulate the generation of amacrine and ganglion cell relative to photoreceptors. A number of potential regulators of regenerative neurogenesis are touched upon, but not further investigated (e.g. *Il1beta*).

Overall, this is a carefully presented analysis that will be useful for a number of fields in which neurogenesis and cell fate decisions are investigated. However, the results are not always stringently bound to the biological model(s) and are overall a bit hard to follow because of that. This limits the broader accessibility and attractiveness of the data presented here.

In particular, a visualisation of the proposed cell fate transition and decision points (as first described from line 59) would aid understanding of the (in principal) strong conceptual framework.

major:

The author state that, different from development, neurons born after injury are not eliminated over time (e.g. in line 82, also line 534f). This is not conclusively shown or referenced. Furthermore, this statement seems to contradict their statement in line 132f that speculates that removal of "excess immature neurons" might be happening.

In the section line 127 - 141 it remains unclear whether NMDA significantly damages photoreceptors or not.

In line 146, no difference in the number of EdU labelled neurons is reported to occur between 7DR and 21DR - could this be a balance between cell proliferation and cell death? What is the evidence for/against this possibility?

The authors state that there they have not found genes for cilia in their expression profile, apart from *FoxJ1a*, which regulates neurogenesis in the retina in the present study. Cultured MGs have been described in mammals (PMID: 25504432), this should be discussed here.

The section around line 169 seems to suggest a replacement of lost neurons between 60 hours post light damage, when ONL shows the most pronounced loss of nuclei and 72 hours. Is 12 hours a reasonable time for the replenishment of ONL nuclei and what is the evidence for that?

Section starting line 332: there's a strong *down*regulation of several genes that are not discussed, e.g. *TNFB*, *tnfsf12*, *stat1a*, while *Il11b* is upregulated at the 36 hour time point - these should be discussed more clearly.

Discussion: Discussions of the results in the first section (page 13) should much more clearly be related back to the model. Schematics clearly illustrating which cell types are lost in each lesion model and which are replaced via which steps are needed to aid interpretation of the results (also would help statements in lines 530 ff).

lines 570ff: the authors speculate, based on their data from the *mmp9* mutant that *mmp9* might control the maturation of *il1b* and that this may mediate some of the observed phenotype. This is

a very testable hypothesis, since the activation of il1b can be inhibited by caspase 1 inhibition.

lines 582: the distinct progenitors described here should be highlighted in the figures.

minor:

line 6: Hitchcock (small i)

line 112: that HAD incorporated

line 115: define "extensive"

line 131: is the "slight decrease" reported here statistically significant?

line 307: "inhibiting GENERATION OF inner retinal neurons"

line 448: "with either uninjected or injected..." there's a grammar problem here

line 508: "that continue to proliferate"

line 529: "amacrines" - jargon

Reviewer #2 (Remarks to the Author):

Lyu et al., 2023

The work by Lyu et al., 2023 provides a significant advancement in our understanding of the molecular mechanisms regulating retina regeneration in zebrafish upon NMDA and light lesions. The authors use a combination of single cell-, single nucleus- and ATAC-sequencing to detect gene expression changes that underlie the two different lesion paradigms, and between regeneration and retinal development. All retinal neurons, with the exception of horizontal cells, are regenerated in both the light lesioned and the NMDA retinae, hence showing that the two paradigms are not as specific as previously thought in terms of ablating distinct retinal neuronal types. Nevertheless, a biased differentiation of MGPCs towards photoreceptors in the light lesioned retina and towards inner neurons in the NMDA-lesioned retina is apparent. Moreover, the authors suggest that matrix metalloprotease Mmp9 may promote generation of inner retinal neurons as compared to photoreceptors. They also point out shared and distinct gene expression between regeneration and development of the retina. Finally, they provide evidence that the transcription factor Foxj1a may be necessary for neuronal regeneration of the adult retina, but not during development.

This reviewer commends the authors for the huge effort to provide this overview of the sc-gene expression changes underlying retina regeneration using several, cutting-edge technologies. Specifically, the manuscript provides a thorough description of the later events of retina regeneration (up to 21 days post-lesion), which have so far been less well described and understood. This comprehensive study of retina regeneration in zebrafish is likely to be of lasting importance in the field, and will support generation of future hypotheses and functional studies in the field. recommends some revision of this manuscript. I have the following comments and suggestions for revisions.

MAJOR COMMENTS

1. Although the manuscript is of a clear and significant importance for the understanding of retina regeneration in zebrafish, the flow of the text and figures is difficult to follow. The readability of the manuscript is likely to improve if the frequent mismatches between the main text and the figures, and with their captions, are corrected.

1a. Specifically, the flow of the text should better follow the order of the figures, to avoid frequent jumps between figure panels. For example, the authors describe first figure 2D, then jump back to describe figure 2C (compare line 208 to line 218). This applies to other figures in the manuscript: first figure S5C/D and then S5A, B, first figure 5E/H and then a jump to figure 6 G,H (lines 413-414).

1b. There are several mismatches between text and related figures, specifically in lines: 229-231; 234-237, 255-258, 258-260, 274-276; 295-295; also, what does the magenta color of the bar plots in figure 3E indicate? The plot looks unclear and difficult to relate to the text. In general, the authors may want to be more precise and exhaustive in their description of figure panels, at least for those mentioned in the highlighted lines. Line 304-figure panel S5H seems to support the opposite of what is stated in this line. There is a severe mismatch between the entire Figure 3 and its caption: panels described in the caption are not matched in the figure, which makes it very hard to understand.

1c. In Figure 1G there is no indication of the uninjured control, whereas in Figure 1J it is not clear what the control for NMDA injections is (uninjected? Vehicle injected?). The uninjured control is also missing from Figure S2.

1d. The nomenclature in Figure 2C does not match the nomenclature in Figure 2D. In Figure 2C the authors list retinal ganglion cells, amacrine cells, bipolar cells, cones, rods, while in Figure 2D the word "precursor" is added to the previous names. The authors should provide a clear definition of what they designate as a "precursor" (see point 2).

1e. Figure panels 4C and 4D are not clearly explained: there is no legend for the colors of the bar plots, as well as for the symbols (what does the circle stand for? And the square? What do E, G, P stand for? Please, specify. The same applies to Figure 6C, 6E, 6F). The way Figure 4F is described is highly unclear to me: lines 338-340. The same applies to figure 6F.

2. The nomenclature used in the manuscript is sometimes confusing. This applies specifically to the terms "resting" Müller glia, "reprogrammed" Müller glia, "activated" Müller glia and "precursor".

2a. How does a reprogrammed MG differ from an activated one? Activated MG is defined in Hoang et al., 2020, but in the present manuscript it seems to be used interchangeably with the term "reprogrammed" MG. How do the two cell populations differ (If they do)?

2b. Please, define what a precursor is in the context of retina regeneration: from the manuscript, it appears that "precursor" defines an immature or differentiating neuron. However, please note that in zebrafish retina development the word precursor refers to specific neuronal progenitors that undergo the last one or two mitoses to generate a specific class of neurons (e.g., horizontal cell precursors described in Godinho et al., 2007; cone precursors described in Suzuki et al., 2013; all reviewed in Amini et al., 2018).

3. Lack of evidence for statements in:

3a. Line 66: is there any published evidence that the excess of regenerated neurons does not integrate into the extant retinal circuit ?

3b. Lines 344-346: it is hard to find the info in the cited table;

3c. Lines 353-354: there is no evidence shown for the implication stated in these lines - maybe move to the discussion section ?

3d. Lines 348-354 significantly interrupt the flow of the text before and after, and the corresponding figure 4G does not fit well with the rest of the panels in figure 4.

4. The authors do not discuss the absence of horizontal cells and their precursors in their lesioned retinae, which are reported however by Lahne et al., 2021 and Celotto et al., 2023. *onecut1*, which is necessary for HC development and expressed strongly in HC precursors (Celotto et al., 2023), appears as a differentially upregulated motif in Figure 2H. How do the authors explain this upregulation, in light of the claimed absence of regenerated HCs and HC precursors upon injury?

5. Figure S7G and line 496: In the image shown for the *foxj1a* mutant, lamination of the retina

looks disrupted, in contrast to what is stated in the text (line: 496). The retinae in the mutant look smaller, and plexiform layers are hardly distinguishable, compared to controls. Also, why is lamination delayed in the *foxj1a* mutant, compared to the control: did the authors check whether the *foxj1a* mutant retina 'catches up' and has developed correctly at time points later than 96 hpf?

6. It is unclear what the birth order of retinal neurons is during development: why do lines 576-577 seem to contradict lines 209-2011?

7. Lines 606-607: Please, provide a different reference for these lines. The cited paper Nagashima et al., 2013 does not resolve the issue of symmetry or asymmetry of the MGPC division. Nagashima et al. showed that Müller glia likely undergo one asymmetric division within 42 hpl with respect to fate, generating a self-renewed MG and a MGPC. They did not examine whether the MGPC itself divides asymmetrically or symmetrically, nor does there appear to be any convincing evidence for an asymmetric division of MGPCs from the published literature.

MINOR COMMENTS

8. Please, kindly revise the font of the gene names throughout the manuscript. The gene names should be indicated in italics, which does not always seem to be the case in the current version of the manuscript. There is a slight typo in the Methods: z stacks are measured as μm and not as μM .

9. Did the authors perform a TUNEL staining to look for signs of unspecific cell death upon light lesion as well as upon NMDA lesion?

10. Please, kindly explain the time points chosen for the NMDA lesion (7DR and 14 DR) and those chosen for the light lesion (7DR, 14 DR and 21 DR) in figure 1A, 1E and 1G. Why did the authors examine also the 21 DR time point in the light lesion, but not in the NMDA lesion? Please, also revise the scheme in Figure 1E: there is no indication of the 21 DR time point.

11. Lines 287-293: The authors may want to expand the description of Figure 3. Please, clearly indicate that HuC/D is a label of RGCs and ACs: this might be obvious for a retina expert, but will be less obvious for readers who are not familiar with the distinct labels of retinal neurons.

12. Line 290: how do you know that they are EdU-positive neurons and not EdU-positive cells?

13. Line 304 appears to be redundant ("in the production of the generation...").

14. Line 306: the authors may want to revise the sentence. The way it is written it suggests that these "initial factors" inhibit the function of inner neurons, whereas they might inhibit the generation of inner neurons.

15. Lines 344-346: this information is not easily accessible in the current table format. In general, I suggest to revise the nomenclature of the tables (ST3, ST4...), because in the current manuscript version each "table" corresponds to an Excel file containing, in fact, several tables.

16. Line 350: Figure S3 is actually a general overview, and does not specifically refer to microglia only. Also, not all the microglia markers listed in the text are visible in the cited Figure S3, only mpeg1.1.

17. It is not clear why the authors sequenced the whole embryo heads (line 366) – please explain.

Reviewer #3 (Remarks to the Author):

In the manuscript, the authors investigated the molecular process of retina regeneration in zebrafish in both light damage and NMDA models at single cell resolution. By comparing the two damage models and normal development process, the authors observed that the regeneration process is similar but distinct between the two damage models and also distinct from normal development process. In addition, the impact of two key factors, Mmp9 and foxj1a, on the regeneration have been examined. The results suggest that Mmp9 plays an important role in repressing regeneration of AC and RGC. In addition, knock down of Foxj1a reduces the number of regenerative neurons. I would like to congratulate the authors for generating such a significant resource and the new insights of the molecular process of retina regeneration in zebrafish. The design of the study is very thorough, and the manuscript is well written. My specific comments are the following:

1. No regeneration of horizontal cells are observed in this study. I am wondering if this is due to no degeneration of horizontal cell in the LD and NMDA damage model.
2. Does Mmp9 affect the normal development of the retina? Is higher AC/RGC to photoreceptor ratio observed in the Mmp9 mutant?
3. It seems that MG activation is faster in the LD model than the NMDA model. Furthermore, given the heterogeneity of MG and derived cells in any given time points, it might be useful to try to take this into account by calculating and correcting pseudotime during DEG and DAR analysis to exclude DEGs between two models due to phase shift.
4. "When examining the MG-to-MGPC branch across the two damage models, noticeable similarities are present. However, unique DEGs and DARs exist between them. The LD model exhibits a heightened neurogenic signal, while the NMDA model emphasizes a more robust inflammatory response pathway. From these observations, the authors suggest a hypothesis: MGPCs from these two damage models are in unique states, leading to varying proportions of retinal neurons being produced. Yet, this difference might merely be a reflection of environmental variations due to different cell type degenerations, rather than distinct MG regeneration pathway's choice. How can we differentiate between these two models?"
5. During typical development, the cell type that a progenitor cell differentiates into is predominantly governed intrinsically. In the context of regeneration, MGPCs differentiate into cell types degenerated. How might MGPCs detect environmental cues and make corresponding differentiation choices? Could the author provide some speculation on this?
6. Based on the data, is foxj1a required for MG activation or subsequent MGPC proliferation and differentiation?
7. Considering the extensive and complex dataset presented, a summarizing model figure highlighting the key findings at the end would be beneficial.

1 We thank the Reviewers for their detailed and constructive comments. We have
2 addressed each individual point raised. Our responses are listed below in blue font:

5 **REVIEWER COMMENTS**

7 Reviewer #1 (Remarks to the Author):

Müller glia in the zebrafish retina respond to injury by generating all retinal cell types.
Different injury paradigms lead to the preferential loss of different retinal cell types. This
offers the unique possibility to assess whether the reaction of Müller glia also
preferentially replaces the lost cell type(s) over the others and which gene
expression/regulatory elements are responsible for any differences. Furthermore,
they address whether regenerative neurogenesis recapitulates developmental
neurogenesis and find it doesn't. The authors present a thorough analysis of the
regulation of gene expression and regulatory elements and discover a number of
interesting functional interactions. Most importantly, mmp9 is shown to regulate the
generation of amacrine and ganglion cell relative to photoreceptors. A number of
potential regulators of regenerative neurogenesis are touched upon, but not further
investigated (e.g. Il1beta).

Overall, this is a carefully presented analysis that will be useful for a number of fields in
which neurogenesis and cell fate decisions are investigated. However, the results are
not always stringently bound to the biological model(s) and are overall a bit hard to
follow because of that. This limits the broader accessibility and attractiveness of the
data presented here.

In particular, a visualisation of the proposed cell fate transition and decision points (as
first described from line 59) would aid understanding of the (in principal) strong
conceptual framework.

We thank the Reviewer for his/her overall positive assessment of the manuscript.

major:

The author state that, different from development, neurons born after injury are not
eliminated over time (e.g. in line 82, also line 534f). This is not conclusively shown or
referenced. Furthermore, this statement seems to contradict their statement in line 132f
that speculates that removal of "excess immature neurons" might be happening.

This conclusion is based on the lack of change in the number of EdU-positive neurons
over time, following labeling from 60-108 hours post-injury, which corresponds closely to
the observed peak in MGPC proliferation observed by both scRNA-Seq and
snRNA/ATAC-Seq analysis (Fig. 2). With one exception, we observe no statistically
significant decrease in EdU incorporation through 14-21 days of recovery post-injury for
any cell type in any injury model, implying that no EdU-positive cells are eliminated

through apoptosis. The one exception is the rather modest reduction in the number of
EdU-labeled rod photoreceptors between 7 and 14 days post-injury, which is referenced
on line 132f. We have modified the text to make this point clearer.

In the section line 127 - 141 it remains unclear whether NMDA significantly damages
photoreceptors or not.

We identify both TUNEL-positive cells and DAPI-positive cells with pyknotic nuclei in the
INL with light damage, as well as in the photoreceptor layer following NMDA (Fig. S3).
This, combined with the corresponding reduction in the number of DAPI-positive nuclei,
supports the conclusion that these cells are dying.

In line 146, no difference in the number of EdU labelled neurons is reported to occur
between 7DR and 21DR - could this be a balance between cell proliferation and cell
death? What is the evidence for/against this possibility?

We have quantified the number of brightly-labeled EdU-positive cells in the section.
EdU was administered between 60 and 108 hours (2.5-4.5 days) post-injury. While
both scRNA-Seq and snRNA/ATAC-Seq showed only very low levels of MGPC
proliferation after this time, a small number of neurons are definitely still being
generated between 7 and 21 days post-injury. The Reviewer is correct that we did not
do EdU labeling between 7-21 days, and while new cells may be generated, the finding
that the number of EdU-labeled cells does not change in this period implies that no
substantial removal of newly-generated neurons is taking place.

The authors state that there they have not found genes for cilia in their expression
profile, apart from FoxJ1a, which regulates neurogenesis in the retina in the present
study. Cultured MGs have been described in mammals (PMID: 25504432), this should
be discussed here.

Muller glia, like virtually every cell type, do indeed possess primary cilia, and we now
cite this reference to emphasize this point. However, Foxj1 and its zebrafish
homologues have been most extensively studied as master transcriptional regulators of
motile multiciliated cells (Stubbs, et al. 2008; Hellman, et al. 2010). Neither Muller glia
nor MGPCs have been observed to be multiciliated, and we likewise do not observe
molecular markers that would suggest that this is the case. This point is now discussed
in more detail in the revised text.

The section around line 169 seems to suggest a replacement of lost neurons between
60 hours post light damage, when ONL shows the most pronounced loss of nuclei and
72 hours. Is 12 hours a reasonable time for the replenishment of ONL nuclei and what is
the evidence for that?

While some of these DAPI-positive nuclei may indeed be regenerated photoreceptors
cells at 72 hours, it is likely that many of them instead represent progenitors undergoing
interkinetic nuclear migration (Lahne and Hyde 2016), in which progenitors transitioning

through the cell cycle migrate apically, divide, and then undergo radial migration in the
basal direction. This is now discussed in the text.

Section starting line 332: there's a strong *down*regulation of several genes that are not
discussed, e.g. TNFb, tnfsf12, stat1a, while Il11b is upregulated at the 36 hour time
point - these should be discussed more clearly.

Tnfb is actually upregulated at 36hrs with similar kinetics to il1b/il11b (Fig. 4G). Tnfsf12
and stat1a are, however, are indeed transiently downregulated following LD at 36hrs,
but upregulated following NMDA injury. This point is now discussed in the text.

Discussion: Discussions of the results in the first section (page 13) should much more
clearly be related back to the model. Schematics clearly illustrating which cell types are
lost in each lesion model and which are replaced via which steps are needed to aid
interpretation of the results (also would help statements in lines 530 ff).

We now include schematic Figure S9 to summarize the overall findings and address
these points.

lines 570ff: the authors speculate, based on their data from the mmp9 mutant that
mmp9 might control the maturation of il1b and that this may mediate some of the
observed phenotype. This is a very testable hypothesis, since the activation of il1b can
be inhibited by caspase 1 inhibition.

This is an excellent suggestion, but we believe it is beyond the scope of the current
study. There are four different Caspase-1 paralogues in zebrafish, and both their
substrate specificity and specificity of known Caspase-1 inhibitors in zebrafish remain
uncertain, so even if partial functional rescue was observed, these results would be
difficult to interpret without further functional analysis of IL-1beta processing and
signaling.

lines 582: the distinct progenitors described here should be highlighted in the figures.

The distinct progenitors described here correspond to postmitotic rod and cone
precursors which has been labeled in red arrow in Figure S4A .

minor:

line 6: Hitchcock (small i)
line 112: that HAD incorporated
line 115: define "extensive"
line 131: is the "slight decrease" reported here statistically significant?
line 307: "inhibiting GENERATION OF inner retinal neurons"
line 448: "with either uninjected or injected..." there's a grammar problem here
line 508: "that continue to proliferate"
line 529: "amacrines" – jargon

These have been corrected.

Reviewer #2 (Remarks to the Author):

Lyu et al., 2023

The work by Lyu et al., 2023 provides a significant advancement in our understanding of
the molecular mechanisms regulating retina regeneration in zebrafish upon NMDA and
light lesions. The authors use a combination of single cell-, single nucleus- and ATAC-
sequencing to detect gene expression changes that underlie the two different lesion
paradigms, and between regeneration and retinal development. All retinal neurons, with
the exception of horizontal cells, are regenerated in both the light lesioned and the
NMDA retinae, hence showing that the two paradigms are not as specific as previously
thought in terms of ablating distinct retinal neuronal types. Nevertheless, a biased
differentiation of MGPCs towards photoreceptors in the light lesioned retina and towards
inner neurons in the NMDA-lesioned retina is apparent. Moreover, the authors suggest
that matrix metalloprotease Mmp9 may promote generation of inner retinal neurons as
compared to photoreceptors. They also point out shared and distinct gene expression
between regeneration and development of the retina. Finally, they provide evidence that
the transcription factor Foxj1a may be necessary for neuronal regeneration of the adult
retina, but not during development.

This reviewer commends the authors for the huge effort to provide this overview of the
sc-gene expression changes underlying retina regeneration using several, cutting-edge
technologies. Specifically, the manuscript provides a thorough description of the later
events of retina regeneration (up to 21 days post-lesion), which have so far been less
well described and understood. This comprehensive study of retina regeneration in
zebrafish is likely to be of lasting importance in the field, and will support generation of
future hypotheses and functional studies in the field. recommends some revision of this
manuscript. I have the following comments and suggestions for revisions.

We thank the Reviewer for his/her positive assessment of the manuscript.

MAJOR COMMENTS

1. Although the manuscript is of a clear and significant importance for the understanding
of retina regeneration in zebrafish, the flow of the text and figures is difficult to follow.
The readability of the manuscript is likely to improve if the frequent mismatches
between the main text and the figures, and with their captions, are corrected.

We thank the Reviewer for his/her careful reading of the text. Please see below:

1a. Specifically, the flow of the text should better follow the order of the figures, to avoid
frequent jumps between figure panels. For example, the authors describe first figure 2D,
then jump back to describe figure 2C (compare line 208 to line 218). This applies to

other figures in the manuscript: first figure S5C/D and then S5A, B, first figure 5E/H and
then a jump to figure 6 G,H (lines 413-414).

We have reorganized the figures to better reflect the order of description.

1b. There are several mismatches between text and related figures, specifically in lines:
229-231; 234-237, 255-258, 258-260, 274-276; 295-295; also, what does the magenta
color of the bar plots in figure 3E indicate? The plot looks unclear and difficult to relate
to the text. In general, the authors may want to be more precise and exhaustive in their
description of figure panels, at least for those mentioned in the highlighted lines. Line
304-figure panel S5H seems to support the opposite of what is stated in this line. There
is a severe mismatch between the entire Figure 3 and its caption: panels described in
the caption are not matched in the figure, which makes it very hard to understand.

We have carefully revised both the figures and legends to greater clarity and to correct
any mismatches. We now clearly state that the magenta bar indicates data from *mmp9*
mutant animals.

1c. In Figure 1G there is no indication of the uninjured control, whereas in Figure 1J it is
not clear what the control for NMDA injections is (uninjected? Vehicle injected?).The
uninjured control is also missing from Figure S2.

Figure 1B represents the PBS-injected, uninjured control for both the light-damage and
NMDA-damage experiments. The purpose of the experiment is to demonstrate how the
number of EdU-labeled cells changes, or not, at different timepoints of recovery from
either LD or NMDA-mediated injury, not compared to the undamaged control. This is
also the case for Fig. S2.

1d. The nomenclature in Figure 2C does not match the nomenclature in Figure 2D. In
Figure 2C the authors list retinal ganglion cells, amacrine cells, bipolar cells, cones,
rods, while in Figure 2D the word “precursor” is added to the previous names. The
authors should provide a clear definition of what they designate as a “precursor” (see
point 2).

A precursor is an immature postmitotic neuron, as opposed to a progenitor, which is
mitotic. This has now been defined explicitly. Figure 2 has been revised for greater
clarity, as requested.

1e. Figure panels 4C and 4D are not clearly explained: there is no legend for the colors
of the bar plots, as well as for the symbols (what does the circle stand for? And the
square? What do E, G, P stand for? Please, specify. The same applies to Figure 6C,
6E, 6F). The way Figure 4F is described is highly unclear to me: lines 338-340. The
same applies to figure 6F.

These figures and the corresponding legends and manuscript text have been revised for
greater clarity, as requested.

2. The nomenclature used in the manuscript is sometimes confusing. This applies
specifically to the terms “resting” Müller glia, “reprogrammed” Müller glia, “activated”
Müller glia and “precursor”.

2a. How does a reprogrammed MG differ from an activated one? Activated MG is
defined in Hoang et al., 2020, but in the present manuscript it seems to be used
interchangeably with the term “reprogrammed” MG. How do the two cell populations
differ (If they do)?

A “reprogrammed MG” in this context represents a Muller glia-derived progenitor cell
(MGPCs). We have altered the text throughout to reflect this.

2b. Please, define what a precursor is in the context of retina regeneration: from the
manuscript, it appears that “precursor” defines an immature or differentiating neuron.
However, please note that in zebrafish retina development the word precursor refers to
specific neuronal progenitors that undergo the last one or two mitoses to generate a
specific class of neurons (e.g., horizontal cell precursors described in Godinho et al.,
2007; cone precursors described in Suzuki et al., 2013; all reviewed in Amini et al.,
2018).

In this context, a precursor is an immature postmitotic neuron, while a progenitor (or
MGPC) is mitotic. This has been defined explicitly in the text.

3. Lack of evidence for statements in:

3a. Line 66: is there any published evidence that the excess of regenerated neurons
does not integrate into the extant retinal circuit ?

There is no direct evidence to this effect. This fact is now stated explicitly.

3b. Lines 344-346: it is hard to find the info in the cited table;

The relevant supplemental dataset (formerly table) has been revised for clarity.

3c. Lines 353-354: there is no evidence shown for the implication stated in these lines -
maybe move to the discussion section ?

This has been done.

3d. Lines 348-354 significantly interrupt the flow of the text before and after, and the
corresponding figure 4G does not fit well with the rest of the panels in figure 4.

We have revised the text for better clarity and smoother narrative flow.

4. The authors do not discuss the absence of horizontal cells and their precursors in

their lesioned retinæ, which are reported however by Lahne et al., 2021 and Celotto et
al., 2023. *oncut1*, which is necessary for HC development and expressed strongly in
HC precursors (Celotto et al., 2023), appears as a differentially upregulated motif in
Figure 2H. How do the authors explain this upregulation, in light of the claimed absence
of regenerated HCs and HC precursors upon injury?

One cut family transcription factors have an essential role in promoting cone
photoreceptor specification in mammals (Lonfat et al. 2021; Emerson et al. 2013),
although this has not been directly shown in zebrafish. Mammalian cone photoreceptors
arise from *One cut/Otx2*-positive neurogenic progenitors that have the potential to
generate either cones or horizontal cells. Interestingly, we observe neurogenic MGPCs
in zebrafish that express both *One cut1/2* and *Otx2*, but do not generate horizontal cells,
raising the possibility that horizontal cell generation might be actively inhibited through
unknown mechanisms in these cells. We discuss this point in the revised manuscript.

5. Figure S7G and line 496: In the image shown for the *foxj1a* mutant, lamination of the
retina looks disrupted, in contrast to what is stated in the text (line: 496). The retinæ in
the mutant look smaller, and plexiform layers are hardly distinguishable, compared to
controls. Also, why is lamination delayed in the *foxj1a* mutant, compared to the control:
did the authors check whether the *foxj1a* mutant retina ‘catches up’ and has developed
correctly at time points later than 96 hpf?

Because the morpholino effect is transient, looking at later timepoints will not be a true
indication of the loss of *foxj1a* expression, as it is likely to increase after the morpholinos
are lost. The size of the embryos was also much smaller in the morphant than the
control and we are unable to separate the issue of the embryo size and the retina size.

More generally, *foxj1a* is broadly expressed in the early embryo, and the morphant
affects the overall size of many organs, including the eye. However, we do not observe
expression of *foxj1a* in retinal progenitors or neural precursors in the developing retina,
and no clear effects on overall levels of retinal neurogenesis in morphants.

6. It is unclear what the birth order of retinal neurons is during development: why do
lines 576-577 seem to contradict lines 209-2011?

This has been clarified in the revised text.

7. Lines 606-607: Please, provide a different reference for these lines. The cited paper
Nagashima et al., 2013 does not resolve the issue of symmetry or asymmetry of the
MGPC division. Nagashima et al. showed that Müller glia likely undergo one asymmetric
division within 42 hpl with respect to fate, generating a self-renewed MG and a MGPC.
They did not examine whether the MGPC itself divides asymmetrically or symmetrically,
nor does there appear to be any convincing evidence for an asymmetric division of
MGPCs from the published literature.

This is absolutely correct. There is no direct evidence to support either symmetric or
asymmetric patterns of cell division by the MGPCs themselves. We revised the text
accordingly.

MINOR COMMENTS

8. Please, kindly revise the font of the gene names throughout the manuscript. The
gene names should be indicated in italics, which does not always seem to be the case
in the current version of the manuscript. There is a slight typo in the Methods: z stacks
are measured as μm and not as μM .

This has been corrected.

9. Did the authors perform a TUNEL staining to look for signs of unspecific cell death
upon light lesion as well as upon NMDA lesion?

As stated in the response to Reviewer 1, we observe pyknotic nuclei in all cell layers
following both LD and NMDA injury. While these often overlap with TUNEL-positive
cells, we observe substantially more pyknotic cells than TUNEL-positive cells. This is
now shown in Figure S3.

10. Please, kindly explain the time points chosen for the NMDA lesion (7DR and 14 DR)
and those chosen for the light lesion (7DR, 14 DR and 21 DR) in figure 1A, 1E and 1G.
Why did the authors examine also the 21 DR time point in the light lesion, but not in the
NMDA lesion? Please, also revise the scheme in Figure 1E: there is no indication of the
21 DR time point.

The 21 DR timepoint was also examined in LD simply because we had more LD-treated
animals available, owing to the fact that this procedure does not involve any direct
manipulation of the animals. In any case, no significant difference is observed between
the 14 DR and 21 DR samples for any of the parameters tested. We have revised the
schematic in Figure 1E to include the 21 DR timepoint.

11. Lines 287-293: The authors may want to expand the description of Figure 3. Please,
clearly indicate that HuC/D is a label of RGCs and ACs: this might be obvious for a
retina expert, but will be less obvious for readers who are not familiar with the distinct
labels of retinal neurons.

This has been done.

12. Line 290: how do you know that they are EdU-positive neurons and not EdU-
positive cells?

As stated in Figures 1 and S1 and S2 we have stained for markers for Muller glia and
microglia, and observe little EdU incorporation, while we observe extensive EdU
incorporation in rods, cones, and HuC/D-positive amacrine and ganglion cells. We
likewise do not observe substantial numbers of any other non-neuronal cell type in our

scRNA-Seq or snRNA/ATAC-Seq analysis. We therefore feel confident in referring to
these as overwhelmingly EdU-positive neurons in this case.

13. Line 304 appears to be redundant (“in the production of the generation...”).

This has been corrected.

14. Line 306: the authors may want to revise the sentence. The way it is written it
suggests that these “initial factors” inhibit the function of inner neurons, whereas they
might inhibit the generation of inner neurons.

This has been corrected.

15. Lines 344-346: this information is not easily accessible in the current table format. In
general, I suggest to revise the nomenclature of the tables (ST3, ST4...), because in the
current manuscript version each “table” corresponds to an Excel file containing, in fact,
several tables.

We now specifically cite these files as Supplemental Datasets rather than Tables to
reduce confusion, and also specifically cite relevant tabs within the datasets in the
following format: Supplemental Dataset X, T(ab) Y.

16. Line 350: Figure S3 is actually a general overview, and does not specifically refer to
microglia only. Also, not all the microglia markers listed in the text are visible in the cited
Figure S3, only mpeg1.1.

We now include a reference to Supplemental Dataset 5, which lists the full complement
of microglial markers.

17. It is not clear why the authors sequenced the whole embryo heads (line 366) –
please explain.

Prior to 36 hpf, it was simply not possible to cleanly dissect retinas. This is now
explained in the text.

Reviewer #3 (Remarks to the Author):

In the manuscript, the authors investigated the molecular process of retina regeneration
in zebrafish in both light damage and NMDA models at single cell resolution. By
comparing the two damage models and normal development process, the authors
observed that the regeneration process is similar but distinct between the two damage
models and also distinct from normal development process. In addition, the impact of
two key factors, Mmp9 and foxj1a, on the regeneration have been examined. The
results suggest that Mmp9 plays an important role in repressing regeneration of AC and

RGC. In addition, knock down of Foxj1a reduces the number of regenerative neurons. I
would like to congratulate the authors for generating such a significant resource and the
new insights of the molecular process of retina regeneration in zebrafish. The design of
the study is very thorough, and the manuscript is well written.

We thank the Reviewer for his/her positive assessment of the manuscript.

My specific comments are the following:

1. No regeneration of horizontal cells are observed in this study. I am wondering if this is
due to no degeneration of horizontal cell in the LD and NMDA damage model.

While we observe no evidence for injury-induced loss of horizontal cells in either injury
model, we lack a cell specific marker to label them, so it is only based on their unique
location (which shifts upon the loss of the ONL) and their morphology. Following injury,
it is often not straightforward to distinguish horizontal cells because of interkinetic
nuclear migration of MGPC nuclei, which enter the OPL and obscure the horizontal
cells. We do not observe evidence for immature MGPC-derived horizontal cell
precursors at any timepoint in either injury model. It is likely that selective ablation of
horizontal cells using techniques such as cell-specific NTR transgenic lines would
indeed lead to selective horizontal cell regeneration. Whether this would also lead to
indirect death and regeneration of photoreceptors and AC/RGC is an interesting topic
for future research.

2. Does Mmp9 affect the normal development of the retina? Is higher AC/RGC to
photoreceptor ratio observed in the Mmp9 mutant?

We now quantify the relative ratio of HuC/D-positive AC/RGC relative to total DAPI-
positive cells and DAPI-positive ONL cells in control samples for both wildtype and
mmp9-deficient fish. We do not observe any changes in the relative ratio of AC/RGC to
photoreceptors. This is included below as Reviewer Figure 1:

**Reviewer Figure 1:** The ratio of the total number of HuC/D and DAPI-positive inner
retinal neurons to the total number of DAPI-positive ONL cells is shown for wildtype and
mmp9-deficient adult retina. Each point represents data from a single animal.

3. It seems that MG activation is faster in the LD model than the NMDA model.
Furthermore, given the heterogeneity of MG and derived cells in any given time points,
it might be useful to try to take this into account by calculating and correcting
pseudotime during DEG and DAR analysis to exclude DEGs between two models due
to phase shift.

We thank the reviewer for this suggestion. In the differential analysis (Figure 2G) of
Rest MG, Act MG, and MGPCs, we did not utilize pseudotime to identify DEGs and
DARs; instead, to eliminate the phase shift effect, we used combined cells from all time
points within a single cell type to call DEGs and DARs. Pseudotime (Figure 2B) was
only employed for visualizing and clustering DEGs and DARs. This methodology is now
detailed in the methods section.

4. "When examining the MG-to-MGPC branch across the two damage models,
noticeable similarities are present. However, unique DEGs and DARs exist between
them. The LD model exhibits a heightened neurogenic signal, while the NMDA model
emphasizes a more robust inflammatory response pathway. From these observations,
the authors suggest a hypothesis: MGPCs from these two damage models are in
unique states, leading to varying proportions of retinal neurons being produced. Yet, this
difference might merely be a reflection of environmental variations due to different cell
type degenerations, rather than distinct MG regeneration pathway's choice. How can we
differentiate between these two models?

It is undoubtedly true that extrinsic factors that are differentially induced by LD vs.
NMDA injury control the formation of fate-biased MGPCs. The text has been revised to
make this point explicitly clear.

5. During typical development, the cell type that a progenitor cell differentiates into is
predominantly governed intrinsically. In the context of regeneration, MGPCs
differentiate into cell types degenerated. How might MGPCs detect environmental cues
and make corresponding differentiation choices? Could the author provide some
speculation on this?

A broad range of extrinsic cues could potentially signal to MGPCs to confer fate biases.
These might include signals differentially released by dying neurons, signals
differentially released from MG or microglia, the loss of contact-dependents signals from
the dying neurons to MG, etc. This is now discussed in the text.

6. Based on the data, is foxj1a required for MG activation or subsequent MGPC
proliferation and differentiation?

This is a good question, and difficult to address without a systematic analysis of
molecular markers specific to both activated MG and MGPCs using techniques such as
scRNA-Seq. We plan to investigate this in future studies.

7. Considering the extensive and complex dataset presented, a summarizing model
figure highlighting the key findings at the end would be beneficial.

A schematic summarizing the findings is now included as Figure S9.

Reviewer #4 comments to the authors:

In the present manuscript, Lyu et al. performed single-nuclear and single-cell RNA-seq and
ATAC-Seq of developing and regenerating retinas to investigate the molecular mechanisms
controlling injury-induced neurogenesis. The results indicated that retinal regeneration was
similar to retinal development, but the regeneration process did not precisely recapitulate retinal
development. The authors also displayed the similarities and differences in gene expression
and gene regulatory networks in both retina damage models. In addition, the authors claimed
that mmp9 was a selective inhibitor of amacrine and ganglion cell formation and foxj1a was
essential for injury-induced neurogenesis. Overall, this study depicted the major differences
between gene regulatory networks between retinal regeneration and development.

Some comments for consideration for the authors are listed below in hope the authors will find
them useful.

1. There are inconsistencies in the content and legends of Figures 1 and 3. For instance, in
Figure 3, the panels C and D in the legend does not present in the actual figure.

This has been corrected.

2. In Figures 1J and 1L, after NMDA and constant light damage, the lowest cell numbers were
seen at 60 hours in the different three layers. Why do different types of injuries lead to the same
results? Can the authors explain the possible reasons for this result?

Both forms of injury are severe, and lead to the loss of the majority of cells directly affected by
the injury in question (photoreceptors following light damage and amacrine/ganglion cells
following NMDA treatment). We do not know why 60 hours represents the peak time for both
direct and indirect neuronal death, although we would predict that if we were to conduct a finer
resolution temporal analysis of cell loss (e.g. sampling every 4 hours between 48 and 72hrs
post-injury), we would observe that the peak of indirect neuronal death (e.g. loss of
amacrine/ganglion cells following light damage) would lag that seen for direct neuronal death.

3. "Both Gene Ontology and KEGG pathway analysis identified functional differences in
differentially expressed genes between the two injury conditions", 1791 genes highly expressed
in the light damage, so how many genes were utilized for this analysis? Did the authors check
whether these genes have statistical significance?

We identified LD-enriched DEGs between LD and NMDA using the following criteria: $\log_{2}fc >$
0.25 and an adjusted p -value < 0.05 . We enumerated the number of DEGs for RestMG, ActMG,
and MGPCs separately in the text and table (Supplemental Dataset 3: T5). Before conducting
GO analysis, we combined the DEGs from RestMG, ActMG, and MGPCs and removed
redundant genes. From this, we obtained 890 and 571 non-overlapping genes enriched in LD or

NMDA, respectively. These genes were re-clustered based on their expression profile along the
trajectory. Subsequently, GO and KEGG analyses were conducted for each cluster of genes.
We've revised the text, tables, and Methods sections to clarify this.

4. For scRNA-seq or multi-omics data, expression of marker genes of individual cell types was
not shown, which make it impossible to validate the cell type annotation.

Marker genes for each cell cluster are shown on the x-axis of Figure S3D, demonstrating the
accuracy of cell type annotation. Also, a list of all the marker genes that are selectively
expressed in each cell type is now included in Supplemental Dataset 5.

5. The authors utilized CCA (canonical correlation analysis) to integrate the single cell RNA-seq
data and single nuclear RNA-seq data, during the process of data integration, did the authors
observe any differences from the two different sequencing methods? Are there variations in the
genes detected through each sequencing approach?

We appreciate the reviewer's question. In this paper, the injury or development samples
from different sequencing methods are processed and analyzed separately. We only
used CCA to integrate the datasets which derived from the same sequencing method
(Figure 2A, 5A, snRNA-seq). Our group has systematically investigated both cell
representation and gene expression patterns in scRNA- and snRNA-Seq datasets
prepared from the same starting material (Santiago et al. 2023). In this study, which
closely reflects our findings in the current study, we found that while both methods can
accurately profile gene expression in major retina cell types, but also observed
differences in cell type proportions captured by snRNA-seq and scRNA-seq.
Specifically, single-cell RNA sequencing overrepresented Müller glia and microglia, but
captured fewer amacrine and retinal ganglion cells (Fig S4A). Cell type-specific scRNA-
Seq profiles generally show higher levels of cross-contamination with transcripts
enriched in other cell types. This is especially so for photoreceptor-specific markers,
which frequently contaminate other cell types in scRNA-Seq datasets, but do so much
less in snRNA-Seq datasets. The number of transcripts and genes detected in scRNA-
Seq samples, as expected, considerably higher than observed with snRNA-seq analysis
of the same cell types. Finally, scRNA-Seq data is significantly enriched for genes
encoding ribosomal proteins, while snRNA-Seq data is enriched for RNA-binding
proteins.

We hope that these changes have addressed any outstanding concerns, and look
forward to hearing your response to the revised manuscript.

**References cited:**

- Emerson, Mark M., Natalia Surzenko, Jillian J. Goetz, Jeffrey Trimarchi, and Constance L. Cepko. 2013.
"Otx2 and Onecut1 Promote the Fates of Cone Photoreceptors and Horizontal Cells and Repress
Rod Photoreceptors." *Developmental Cell* 26 (1): 59–72.
Lahne, Manuela, and David R. Hyde. 2016. "Interkinetic Nuclear Migration in the Regenerating Retina."
*Advances in Experimental Medicine and Biology* 854: 587–93.
Lonfat, Nicolas, Su Wang, Changhee Lee, Mauricio Garcia, Jiho Choi, Peter J. Park, and Connie Cepko.

2021. "Cis-Regulatory Dissection of Cone Development Reveals a Broad Role for Otx2 and Oc
Transcription Factors." *Development* 148 (9). <https://doi.org/10.1242/dev.198549>.
Santiago, Clayton P., Megan Y. Gimmen, Yuchen Lu, Minda M. McNally, Leighton H. Duncan, Tyler J.
Creamer, Linda D. Orzolek, Seth Blackshaw, and Mandeep S. Singh. 2023. "Comparative Analysis of
Single-Cell and Single-Nucleus RNA-Sequencing in a Rabbit Model of Retinal Detachment-Related
Proliferative Vitreoretinopathy." *Ophthalmology Science* 3 (4): 100335.

REVIEWERS' COMMENTS

Reviewer #1 (Remarks to the Author):

The authors have carefully considered my comments and answered most. The added specifications, discussions and figure S9 significantly aid the understanding of the complex results.

I still think an experimental validation of the II1beta hypothesis would have strengthened the manuscript, but I am looking forward to these results in a follow up manuscript.

Reviewer #2 (Remarks to the Author):

The authors have substantially revised the ms along the lines suggested by the reviewers. They have convincingly addressed all the points that I have raised and fixed the various smaller issues, and the revised version of the ms is therefore now fine in my opinion. Congratulations on a nice piece of work !

Reviewer #3 (Remarks to the Author):

The author had adequately addressed issues and questions raised by the reviewers by modifying/correcting the text and add additional supplement figures. The study is well designed and clearly written. I believe the manuscript is ready to be accepted for publications.

Reviewer #4 (Remarks to the Author):

The authors addressed all my concerns, no further question.